# Capsular specificity in temperate phages of *Klebsiella pneumoniae* is driven by diverse receptor-binding enzymes

Aleksandra Otwinowska[1]☯, Janusz Koszucki[2,3]☯, Vyshakh R. Panicker[2,3], Jade Leconte[2], Sebastian Olejniczak[1], Kathryn E. Holt[4,5], Edward J. Feil[6], Eduardo P. C. Rocha[7], Bogna Smug[2], Barbara Maciejewska[1], Zuzanna Drulis-Kawa[1]*, Rafal J. Mostowy[2]*

**1** Department of Pathogen Biology and Immunology, University of Wroclaw, Wrocław, Poland, **2** Malopolska Centre of Biotechnology, Jagiellonian University, Kraków, Poland, **3** Doctoral School of Exact and Natural Sciences, Jagiellonian University, Kraków, Poland, **4** Department of Infection Biology, London School of Hygiene & Tropical Medicine, London, United Kingdom, **5** Department of Infectious Diseases, School of Translational Medicine, Monash University, Melbourne, Victoria, Australia, **6** The Milner Centre for Evolution, Department of Life Sciences, University of Bath, Bath, United Kingdom, **7** Institut Pasteur, Université Paris Cité, CNRS, UMR3525, Microbial Evolutionary Genomics, Paris, France

☯ These authors contributed equally to this work.
* rafal.mostowy@uj.edu.pl (RJM); zuzanna.drulis-kawa@uwr.edu.pl (ZD-K)

## Abstract

Virulent bacteriophages infecting *Klebsiella pneumoniae often show capsule-driven host tropism* due to the presence of capsule-specific depolymerases. Yet for temperate phages the genetic and functional basis of such capsular specificity remains less well understood. Depolymerases appear unexpectedly rare in prophage genomes, raising unresolved questions about which prophage genes mediate capsular specificity, whether this apparent scarcity reflects biological or ecological differences versus annotation limitation, and whether prophage-encoded receptor-binding proteins (RBPs) are functionally active. To address these questions, we analysed 3,900 *Klebsiella* genomes from diverse ecological niches to identify prophage-encoded proteins mediating capsular specificity. We conducted a genome-wide association study (GWAS) correlating prophage protein clusters (from 8,105 prophages) with confidently assigned bacterial K-loci. GWAS revealed statistically supported predictors of capsular specificity for 16 of the 35 most diverse K-loci analysed. These predictors were dominated by diverse RBPs, including classical $\beta$-helix depolymerases (6 predictors), SGNH-domain hydrolases predicted to deacetylate polysaccharides (6 predictors), and structurally novel RBPs lacking known depolymerase folds (2 predictors). Nearly one-third of K-loci yielded no statistically significant predictors. A targeted experimental screen of 50 candidate prophage depolymerases showed that 34 failed to yield detectable recombinant expression, and neither sequence similarity, structural prediction, nor prophage genomic context reliably predicted activity. Of the 14 active enzymes, 5 targeted a K-type different from that predicted of their bacterial

**Data availability statement:** All input data and intermediate files generated by the workflow and required to reproduce the results, but not provided as Supporting Information, are publicly available in Figshare under the following DOI: - https://doi.org/10.6084/m9.figshare.29181188. These include genome assemblies for 3,911 Klebsiella isolates (whole-genome nucleotide sequences, CDS and predicted protein FASTA files), corresponding GenBank annotation files, isolate metadata tables, contig information, and the IQ-TREE bacterial phylogeny in Newick format. The repository further contains sequences and annotations for 8,105 detected prophages (genome, CDS and protein FASTA files and GenBank files), together with prophage metadata, protein cluster assignments (including representative sequences) and functional annotation tables derived from HH-suite analyses. All analysis scripts and workflows are archived in Zenodo: - Analysis pipeline (Zenodo archive): https://doi.org/10.5281/zenodo.18682073 - Figure generation pipeline (Zenodo archive): https://doi.org/10.5281/zenodo.18699826 For convenience, the corresponding GitHub repositories are also available at: https://github.com/bioinf-mcb/mgg_gwas https://github.com/rmostowy/klebs-gwas-figs.

**Funding:** This work was supported by the Polskie Powroty programme of the Narodowa Agencja Wymiany Akademickiej (NAWA; https://nawa.gov.pl/en) awarded to RM, an Installation Grant from the European Molecular Biology Organisation (EMBO; https://www.embo.org) awarded to RM, and grant no. 2020/38/E/NZ8/00432 from the Narodowe Centrum Nauki (NCN; https://www.ncn.gov.pl/en) awarded to RM. Support from the INCEPTION project (PIA/ANR-16-CONV-0005) awarded to EPCR and the Laboratoire d'Excellence IBEID Integrative Biology of Emerging Infectious Diseases (ANR-10-LABX-62-IBEID) awarded to EPCR is also acknowledged. Italian strains were collected as part of the SpARK project funded under the 2016 Joint Programming Initiative on Antimicrobial Resistance call 'Transmission dynamics' (Medical Research Council (MRC) reference no. MR/R00241X/1) awarded to EJF. We gratefully acknowledge the Polish high-performance computing infrastructure PLGrid (https://plgrid.pl/; HPC centres: CI TASK

host, and enzyme specificity was not consistently explained by sequence or structural homology. Comparison with GWAS predictions revealed that 10 of the 12 strongest GWAS predictors were experimentally validated, while 2 remained inconclusive. Together, these results highlight the intrinsic difficulty of predicting activity and capsular specificity of prophage-encoded RBPs from genomic information alone. Finally, analysis of 4,598 high-completeness prophages revealed that SGNH-domain hydrolases are among the most prevalent enzymatic domains in prophage RBPs. Two SGNH-domain RBPs identified by GWAS were experimentally confirmed as active esterases, supporting capsule deacetylation as a widespread alternative to polysaccharide depolymerisation in temperate phages. Our findings reveal that *Klebsiella* prophages encode structurally diverse RBPs, suggesting temperate phages may rely not only on depolymerisation but also on capsule modification–such as deacetylation–for infection. This also suggests that capsule modification may contribute to phage–host interactions in ways not fully captured by current K-locus assignments, with potential implications for phage specificity, competition and vaccine design.

## Introduction

Host range is among the most critical traits of viruses, determining the variety of hosts that can be infected by a given virus [1]. Understanding and predicting viral host range is fundamental for studying viral evolution, ecology and infection dynamics [2]. In bacteriophages, host range is driven by co-evolution with bacterial hosts [3,4] and shaped by horizontal gene transfer [5,6], with broad relevance for phage therapy and microbiome engineering [7–10]. However, accurate prediction of viral host range remains challenging due to the complexity of virus-host interactions, from initial adsorption to successful evasion of host defence systems [11]. Nevertheless, recent advances in multi-omics and computational biology have enabled statistical predictions that can be validated empirically [12]. Indeed, two recent genomic studies demonstrate that such approaches can help successfully predict the infectivity of virulent phages in two species of *Enterobacteriaceae*, *Klebsiella pneumoniae* [13] and *Escherichia coli* [14], underscoring the critical role of receptor-binding proteins (RBPs) in determining virus host range.

*K. pneumoniae* in particular serves as a powerful system for studying bacteria-virus interactions. It is an opportunistic bacterial pathogen characterised by the extensive diversity of its capsular polysaccharides (CPS) [15]. This diversity encompasses 77 serologically defined CPS types (K-types) with elucidated repeat-unit structures [16], each corresponding to a characterised capsule synthesis locus (K-locus), and at least 86 additional genetically distinct K-loci identified through genome-based typing [17–19]. Although most KL-types lack resolved CPS structures, several have now been resolved [20–24]. Experimentally isolated virulent phages infecting *K. pneumoniae* typically produce visible halos on bacterial lawns, indicating enzymatic degradation of CPS [25]. These halos result from phage RBPs with depolymerase activity, enabling recognition, degradation, and subsequent infection [26]. Such enzymes

and ACK Cyfronet AGH) for providing computational resources and support under grants no. PLG/2023/016559 and PLG/2024/017016 awarded to RM. The funders had no role in study design, data collection and analysis, decision to publish, or preparation of the manuscript.

**Competing interests:** The authors have declared that no competing interests exist.

**Abbreviations:** CIP, Collection de l'Institut Pasteur; CPS, capsular polysaccharides; ECODs, Evolutionary Classification of Protein Domains; FN, false negative; FP, false positive; GWAS, genome-wide association study; KASPAH, *Klebsiella* Acquisition Surveillance Project at Alfred Health; KPSC, *K. pneumoniae* Species Complex; MCC, Matthews Correlation Coefficient; MHFC, minimal halo-forming concentration; NCTC, National Collection of Type Cultures; ONT, Oxford Nanopore Technology; ORFs, Open Reading Frames; pNPA, *p*-nitrophenyl acetate; RBPs, receptor-binding proteins; SDS-PAGE, sodium dodecyl sulfate–polyacrylamide gel electrophoresis; SCs, sequence clusters; TSB or TSA, Tryptone Soya Broth or Agar; TN, true negative; TP, true positive; UKHSA, UK Health Security Agency.

generally exhibit highly specific activity against individual CPS types, with 58 experimentally validated depolymerases described so far, targeting 31 distinct K-types [27]. Structurally, these depolymerase-containing RBPs are typically homotrimers, featuring a characteristic right-handed parallel $\beta$-helix [28–31]. As a result, the capsular specificity of these depolymerase-containing RBPs is thought to be a primary determinant of phage host range in *K. pneumoniae*, giving rise to capsule-driven host tropism. Nevertheless, our understanding of the genetic, structural diversity and specificity of depolymerase-containing RBPs in *Klebsiella* phages remains far from complete.

Given that our knowledge about phage depolymerases comes predominantly from studies focused on virulent phages, their role in the biology of temperate phages remains even less understood. In *K. pneumoniae*, studies have demonstrated capsule-driven host tropism in temperate phages, observing that prophages induced from bacterial genomes could infect only bacterial lineages with identical K-loci, evidenced by characteristic halos indicative of depolymerase activity [32]. Yet, attempts to computationally identify corresponding depolymerase genes in prophages failed to show clear correlations with the observed infection patterns. Subsequent genomic analyses of nearly 4,000 *K. pneumoniae* genomes have similarly reported that, while serotype (K-type) specificity predicts phage-mediated gene flow between bacterial lineages, identifiable depolymerase sequences in prophages remain surprisingly scarce [33]. More recently, a machine-learning study showed that genetic features of prophage-encoded depolymerases could partially predict both the K-locus of the bacterial host and the host range of virulent phages [34]. However, the predictive power of these models was generally low, and depolymerases were detected in only about one-fifth of prophages, raising the question of whether this apparent scarcity reflects technical limitations in annotation or fundamental biological differences between virulent and temperate phages.

In light of these observations, several key gaps remain in our understanding of capsule-driven host tropism in temperate *Klebsiella* phages. First, we still do not know which prophage genes determine capsular specificity, or whether these determinants correspond to the canonical depolymerases that mediate host range in virulent phages. Second, depolymerase genes appear strikingly rare in prophage genomes, and it is unclear whether this scarcity reflects a genuine biological difference–such as alternative infection strategies–or simply limitations in our ability to functionally annotate divergent receptor-binding enzymes. Third, even when putative depolymerases can be identified, it remains unknown whether they are active against the capsule type of their host, or any other K-type, given that capsule switching provides a common route for escaping phage infection. Addressing these questions is essential to understand how temperate phages interact with capsulated bacteria and to predict the host range encoded within prophage genomes.

To address these gaps, we aimed to identify genetic determinants of capsular specificity in temperate phages of *K. pneumoniae* using a Genome-Wide Association Study (GWAS) approach. Due to the extensive diversity of bacterial polysaccharides in *K. pneumoniae*, we did not aim to identify a single universal determinant of

capsular specificity. Rather, we sought to identify recurrent prophage-encoded protein families statistically associated with specific K-loci, and to test representative candidates experimentally. GWAS is the right tool for this task as it identifies genetic variants statistically associated with phenotypes of interest, an approach already successfully adapted to bacterial genomics [35–38]. Compared to other statistical learning methods, GWAS has the advantage of not relying on pre-labelled training datasets, thereby avoiding predefined functional expectations and allowing for biologically interpretable associations between genetic variants and phenotypic traits. We applied GWAS to a diverse dataset of *K. pneumoniae* genomes from two systematically curated collections (Australia [39] and Italy [40]), representing near-random sampling from a collection of 3,900 genomes across diverse ecological and clinical niches. Within this dataset, we predicted prophages and employed GWAS on the subset of 2,527 genomes for which we could confidently predict the type of capsular polysaccharide (K-type), identifying prophage protein clusters statistically associated with 35 common K-types as candidate predictors of capsular specificity. Independently, we manually curated depolymerase-containing RBPs from prophages found in 99 representative bacterial genomes (KASPAH-REF) and performed recombinant protein preparation and enzymatic activity testing against 118 K-type strain collection, using 58 previously characterised depolymerases from virulent phages as a comparative reference [27] (see S1 Table; see also Fig 1 for a visual summary of our approach). Our results demonstrate that while prophages can be a rich source of structurally and functionally diverse depolymerases, they are also an unpredictable one. Our approach also uncovers previously underappreciated classes of receptor-binding proteins dominated by SGNH-domain enzymes, which are likely involved in removing capsule modifications such as acetyl groups, and pointing to capsule deacetylation as a widespread alternative mechanism of infection initiation.

## Results

The aim of this study was 3-fold: (a) to identify genetic determinants of capsular specificity in temperate phages of *K. pneumoniae*, (b) to functionally characterise putative depolymerase candidates from a subset of isolates, and (c) to compare genomic predictions with experimental outcomes. To this end, we carried out a comprehensive search that is described in detail in Methods and visually summarised in Fig 1. Briefly, we assembled a dataset of 3,900 *Klebsiella* genomes by combining two large and systematically curated sources. The KASPAH dataset includes 391 clinical and colonisation (gut/throat) isolates collected from hospital patients in Melbourne [39,41,42], while KLEBPAVIA comprises 3,509 isolates sampled through a One Health framework in northern Italy, spanning human, animal, environmental, and agricultural sources [40]. Together, these datasets provide a comprehensive and ecologically broad view of *Klebsiella* diversity, avoiding the clone- and outbreak-driven biases often seen in public genome databases. All isolates were sequenced using a short-read technology, and 99 representative KASPAH genomes were additionally sequenced using long reads to produce high-quality hybrid assemblies [43] (KASPAH-REF). For each genome, we identified the capsular locus (K-locus) using Kaptive [18] and assigned bacterial isolates into sequence clusters (SCs) using PopPunk [44]; the latter was done to account for population structure in the GWAS analysis, minimising confounding from recent shared ancestry and prophage co-inheritance among closely related isolates.

Prophage regions ($n = 8,105$) were identified using VirSorter [45], PhiSpy [46], and CheckV [47], retaining both high-completeness (likely intact) and medium-completeness predictions to capture fragmented prophages in short-read assemblies (see S1 Fig for a distribution of prophage completeness in most frequent K-types). To facilitate robust comparisons, we focused subsequent analyses on 2,527 genomes within the *K. pneumoniae* species complex (KPSC) that had confident K-locus predictions, and focused on 35 K-loci with sufficient diversity ($\geqslant 10$ SCs) to ensure statistical power in subsequent analyses. This dataset is visualised in Fig 2, with the bacterial phylogeny shown in Fig 2A and bacterial and prophage diversity across K-loci shown in Fig 2B. The 35 K-loci each represented between 10–24 SCs and from 29 phage variants for KL47–112 phage variants for KL30 (median of 55), hence encompassing considerable diversity in both capsule type and prophage content. The full distribution of SCs and phage variants per K-locus is shown in S2 Fig. The bacterial and prophage metadata are provided in S1 and S2 Data, respectively.

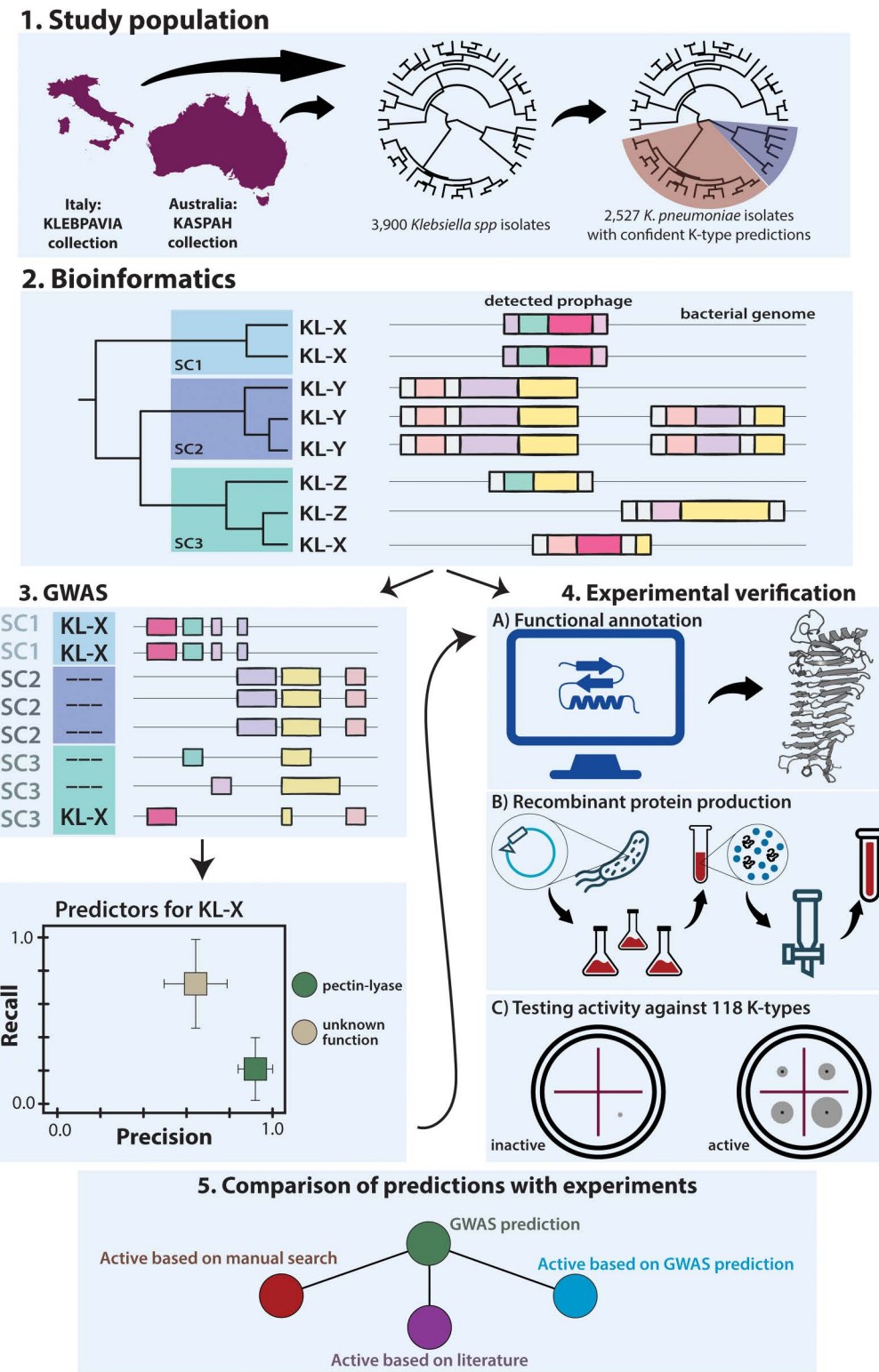

**Fig 1. Visual summary of the study approach.** (1) We assembled an ecologically diverse collection of 3,900 *Klebsiella* genomes, of which 2,527 *Klebsiella pneumoniae* isolates had high-confidence capsular (K-locus) assignments. (2) Prophage regions were predicted in each genome, and all prophage-encoded proteins were clustered and functionally annotated. (3) A bacterial GWAS (elastic-net model) was then performed on 2,527 isolates to

link prophage protein clusters to 35 most diverse K-loci, yielding protein clusters as candidate predictors of capsular specificity. (4) 50 manually selected putative depolymerases from 99 high-quality reference genomes ("KASPAH-REF") were selected for recombinant protein production and enzymatic activity testing against a panel of 118 *Klebsiella* strains with diverse K-types. (5) GWAS predictors (green node) were compared to these experimental results (red node) and to previously characterized virulent-phage depolymerases (purple node); for predictors lacking any known sequence similarity, we selected representative $\beta$-helix (pectin-lyase domain) proteins for recombinant production and functional testing (blue node). Country outlines were obtained from Natural Earth (Admin 0 – Countries, 1:10m scale; https://www.naturalearthdata.com/downloads/10m-cultural-vectors/).

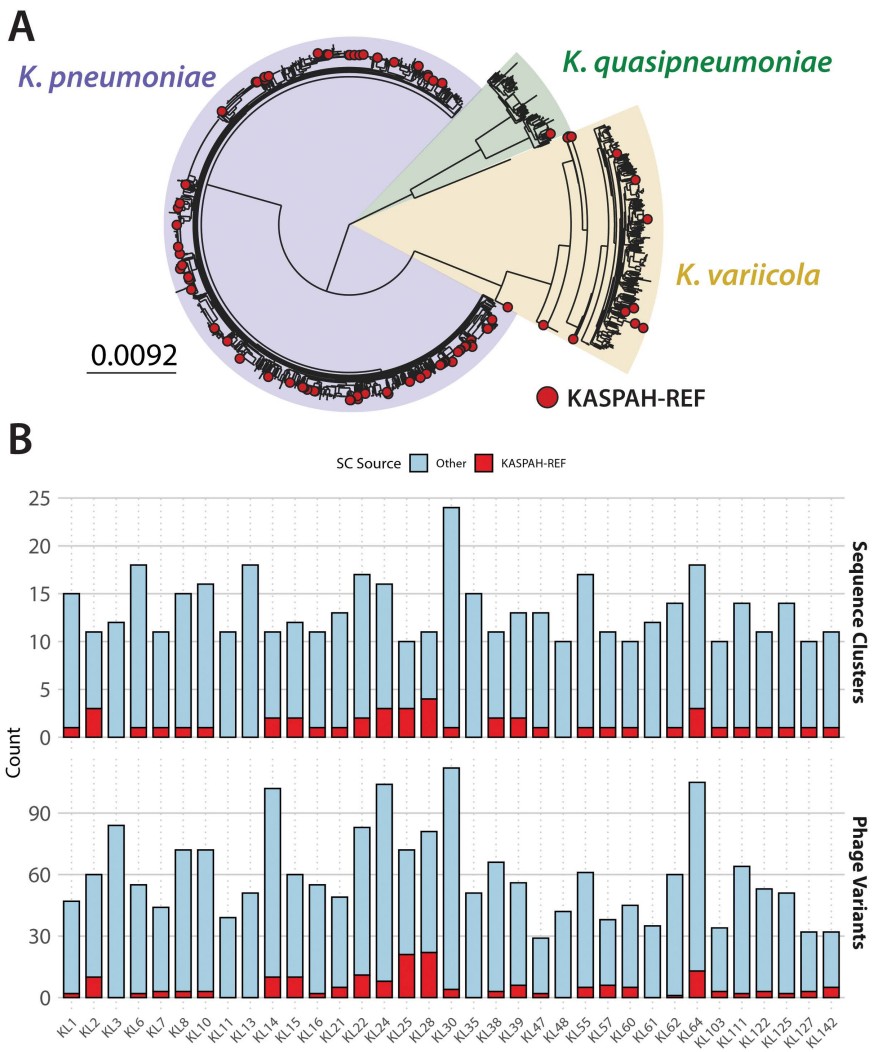

**Fig 2. The dataset. (A)** Maximum-likelihood tree of 2,527 isolates from the *K. pneumoniae* Species Complex (KPSC). Red dots show isolates belonging to the KASPAH-REF subset ($n = 99$). Shaded areas show the three main species from the KPSC with reliable K-locus predictions: *K. pneumoniae* (purple), *K. variicola* (yellow), and *K. quasipneumoniae* (green). **(B)** Top plot shows bacterial diversity as the number of sequence clusters (SCs) per predicted K-locus amongst KASPAH-REF isolates (red) and all other isolates (blue). Bottom plot shows phage diversity measured by the number of phage variants (similar prophages clustered using the wGRR = 0.95 threshold; see Methods) identified in the KASPAH-REF isolates (red) and using all 2,428 remaining isolates (blue). The plot includes only the 35 most diverse K-loci in our dataset, defined as those present in at least 10 different SCs; these 35 K-loci were selected for GWAS analyses. The data and phylogeny underlying this Figure can be found at Figshare (https://doi.org/10.6084/m9.figshare.29181188), S1 Data, S2 Data, and can be reproduced using code archived in Zenodo (https://doi.org/10.5281/zenodo.18699826).

PLOS Biology

## GWAS predicts structurally diverse receptor-binding proteins

To perform a genome-wide association study (GWAS), we analysed prophage protein clusters as genetic predictors of capsular specificity by correlating their presence with K-locus assignments. Here, *genetic predictors* of capsular specificity refer to statistically supported associations between prophage-encoded proteins and bacterial K-loci, which we hypothesised would reflect biologically meaningful capsule-recognition mechanisms. To this end, we used an elastic-net regression model– a machine-learning extension of linear regression designed for genomic settings with many correlated predictors. In our case, the predictors are the presence or absence of each prophage protein cluster in an isolate, and the response variable is the presence or absence of a given K-locus. Because multiple protein clusters can co-occur within prophages and many are homologous, this results in a high-dimensional predictor matrix with substantial correlation structure. Elastic-net accommodates this by combining lasso, which selects a small number of informative clusters by shrinking others to zero, with ridge regression, which retains groups of correlated predictors by shrinking their coefficients together. This allows the model to detect a single dominant capsule-specific RBP when one exists, while also capturing cases where several related or mosaic proteins correlate with the same K-type – a behaviour that matches the biological diversity of phage RBPs. To minimise confounding from lineage effects, population structure was incorporated by including sequence clusters (SCs) as covariates in the regression model. The elastic-net procedure then estimates a regression coefficient ($\beta$) for each protein cluster, indicating the strength and direction of its association with the given K-locus. Predictors were retained when $\beta > 0$ and the p-value for that coefficient was below 0.05 (following Bonferroni-correction for multiple testing).

To evaluate the performance of elastic-net predictors, we used standard classification metrics based on true and false predictions. In our setting, a true positive (TP) is a protein cluster present in an isolate with the corresponding K-locus, a false positive (FP) is a protein cluster present in an isolate with a different K-locus, and a false negative (FN) is a protein cluster absent from an isolate with the corresponding K-locus. Precision (TP/(TP+FP)) measures the proportion of predictor occurrences that correctly identify their associated K-locus, whereas recall (TP/(TP+FN)) measures the proportion of isolates with a given K-locus in which the corresponding predictor is detected. The F1 score combines precision and recall into a single harmonic mean, favouring predictors that balance both metrics. The Matthews Correlation Coefficient (MCC) incorporates all four quantities (TP, FP, FN, true negatives), providing a more robust measure of overall classification performance, especially for imbalanced datasets such as ours. Full details of the GWAS methodology are provided in Methods.

Fig 3A shows the results of GWAS for predictors filtered with F1 $\geqslant$ 0.5 and MCC $\geqslant$ 0.5. Using those thresholds, we obtained 35 protein clusters associated with 24 out of 35 K-loci, with eight K-loci (KL3, KL11, KL14, KL21, KL24, KL25, KL111 and KL127) having multiple associated predictors. Functional annotation of these 35 clusters using PHROGs [48] revealed that 23 of them encoded tail-associated proteins, of which 18 were predicted to be tail fibres–consistent with our expectation that capsule-specific associations would predominantly involve receptor-binding proteins (RBPs). We also observed a negative correlation between precision and recall across predictors (Pearson's *r* = −0.58, 95% CI: −0.76 to −0.30; see S3 Fig). While such a trade-off is expected, the presence of low-precision, high-recall predictors suggests that some candidates may target multiple K-types or represent associations unrelated to capsule interaction (more on this later).

Given that the initial GWAS results predominantly identified receptor-binding proteins (RBPs), we next focused on refining predictors based on biological expectations of polysaccharide specificity. Since most known depolymerases act on a single capsule type [27], we applied a stringent precision threshold ($\geqslant$ 0.8) to prioritise highly specific predictors. Furthermore, since RBPs are often modular, with variation across N-terminal and C-terminal domains [49], we reasoned that the clustering thresholds used to define protein families could affect whether related proteins were grouped into single predictors or split into distinct predictors. To address this, we systematically repeated GWAS across six clustering thresholds, combining identity cutoffs (none, 50%, 80%) with coverage thresholds (50%, 80%), maximising chances that

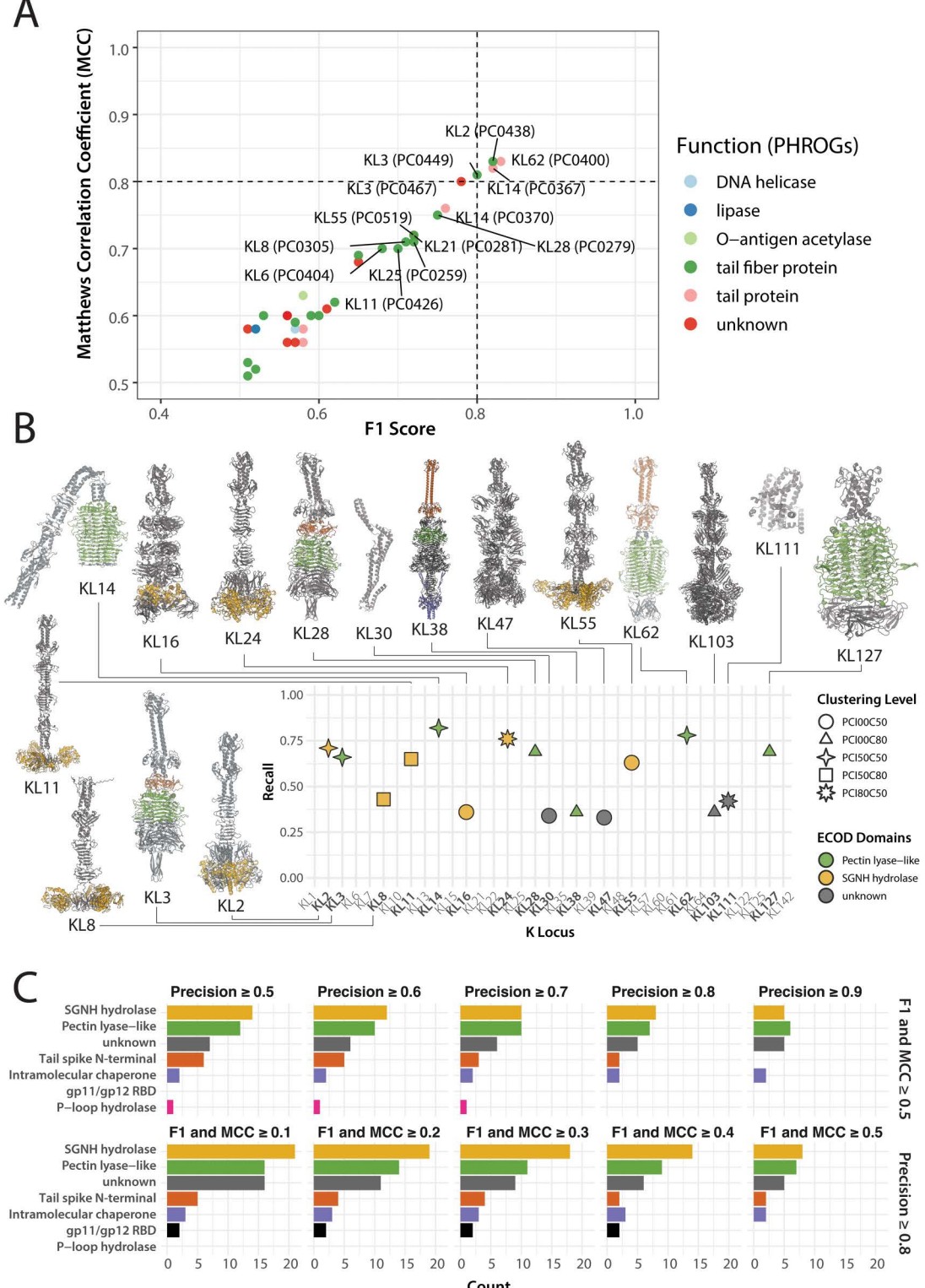

**Fig 3. GWAS predicts diverse receptor-binding proteins. (A)** F1 score vs. Matthews Correlation Coefficient (MCC) for prophage protein clusters (PC50 clustering) identified as GWAS predictors with F1 ⩾ 0.5 and MCC ⩾ 0.5. Colours indicate functional annotations assigned using PHROGs. **(B)** Recall of the strongest predictor identified for each K-locus. For each K-locus, predictors with precision ⩾ 0.8 were selected, retaining the one with the

highest F1×MCC product across clustering thresholds; only those with F1 ⩾ 0.5 and MCC ⩾ 0.5 are shown (with the corresponding K-loci names bolded). Shapes indicate the clustering threshold at which the strongest predictor was identified; colours reflect the top predicted ECOD domain, or grey for unannotated regions. For the 16 predictors obtained, AlphaFold3 models were generated based on representative proteins, with ECOD domains coloured based on HHpred predictions. **(C)** Distribution of ECOD domains across GWAS predictors across different filtering thresholds, shown as bar plots. The top row summarises predictors meeting F1 ⩾ 0.5 and MCC ⩾ 0.5, across varying precision thresholds (0.5–0.9). The bottom row shows predictors filtered at minimum precision ⩾ 0.8, across varying F1 and MCC thresholds (0.1–0.5). ECOD domain names are abbreviated as follows: *SGNH hydrolase* (yellow; T-level: 2007.5.1), *Pectin-lyase like* (green; T-level: 207.2.1), Tail spike N-terminal (brown; Full name: *Putative tailspike protein Orf210 N-terminal domain*, T-level: 3856.1.1), Intramolecular chaperone (purple; Full name: *Intramolecular chaperone domain in virus tail spike protein*, T-level: 3240.1.1), gp11/gp12 RBD (black; Full name: *gp11/gp12 receptor-binding domain*, T-level: 877.1.1), P-loop hydrolase (pink; Full name: *P-loop containing nucleoside triphosphate hydrolases*, T-level: 2004.1.1). Grey denotes the absence of functional hits. The data underlying this Figure can be found at Figshare (https://doi.org/10.6084/m9.figshare.29181188), and can be reproduced using code archived in Zenodo (https://doi.org/10.5281/zenodo.18699826).

such potential modularity was captured. For each K-locus, we then selected the clustering level that maximised predictor performance (defined as the highest product of F1 score and MCC, with both metrics ⩾ 0.5), aiming to identify the most biologically informative protein cluster for each of them.

Using these criteria, we obtained predictions for 16 out of 35 K-loci (Fig 3B). We then selected a representative protein from the best predictor (protein cluster) and generated its 3D models using AlphaFold3 [50]. (Note that for 19 K-loci, there were no predictors that fulfilled our criteria; see Discussion.) We also mapped the locations of the major ECODs (Evolutionary Classification of Protein Domains) [51] detected in those proteins onto the 3D structures (see Fig 3B). We found that 6 of those 16 proteins (from predictors for KL3, KL14, KL28, KL38, KL62, KL127) were classical depolymerases with the characteristic, parallel $\beta$-helix fold (ECOD T-level: 207.2.1, *Pectin-lyase like*). Another 6 proteins (from predictors for KL2, KL8, KL11, KL16, KL24 and KL55) lacked the parallel $\beta$-helix, carrying a *SGNH hydrolase* domain (ECOD T-level: 2007.5.1), and four remaining proteins had no significant ECOD hits identified. Of those four, two proteins (predictors for KL47 and KL103) looked like unusual RBPs with previously uncharacterised folds, while the remaining two (predictors for KL30 and KL111) did not look like RBPs. We used FoldSeek [52] to search the non-RBP predictors against the CATH Protein Structure Classification database [53], the Protein Data Bank database [54], and the AlphaFold Protein Structure database [55] and found that these proteins structurally resembled known acetyltransferases.

Finally, we examined how varying threshold for filtering GWAS predictors influenced the distribution of ECOD domains identified among the associated protein clusters. Regardless of the filters used, filtered predictors were dominated by two structural folds: the *Pectin lyase-like* domain and the *SGNH hydrolase* domain, along with a substantial fraction of proteins lacking functional annotation (Fig 3C). These results suggest that capsular specificity in temperate phages of *K. pneumoniae* may rely not only on classical depolymerases but also on other structurally diverse RBPs with distinct folds. Furthermore, we observed that by relaxing predictor filtering criteria – specifically, allowing lower recall while maintaining high precision – additional lower-frequency protein clusters emerged, some of which may encode previously uncharacterised $\beta$-helix-containing depolymerases. We therefore next lowered the filtering criteria and applied a series of manual curation steps to identify putative depolymerases against as many K-loci as possible.

## Prophages are an unpredictable source of active depolymerases

To comprehensively identify GWAS-predicted depolymerases containing the characteristic right-handed $\beta$-helix fold, we developed a semi-supervised curation framework combining multiple evaluation steps (full details in S1 Text). Starting from statistically significant GWAS predictors, we first assessed precision and recall for each protein cluster at both isolate and sequence-cluster (SC) levels to account for population structure. To accommodate the extensive modularity of phage receptor-binding proteins (RBPs), we then evaluated six alternative clustering thresholds (coverage thresholds of 50% and 80%; sequence identity thresholds of 0%, 50%, and 80%) to identify parameter settings that best captured biologically coherent protein families for each K-locus. Next, we applied phylogenetic mapping using Phandango [56] to visualise whether predictors consistently tracked with K-locus distribution across the bacterial tree. Finally, based on combined

evidence from classification metrics, clustering behaviour, domain annotation and structural modeling, predictors were conservatively classified as 'strong' or 'likely'. Using this approach, we identified 12 'strong' predictors for 10 K-loci (KL3, KL10, KL14, KL25, KL28, KL38, KL60, KL62, KL64, KL111) and 14 'likely' predictors for 11 K-loci (KL7, KL14, KL15, KL22, KL24, KL25, KL30, KL47, KL64, KL122, KL127), together spanning 18 of the 35 analysed K-loci (Fig 4A). All identified predictors, together with their representative protein sequences and associated metadata, are provided in S3 Table.

Separately, we experimentally screened prophage-encoded candidate depolymerases from the KASPAH-REF dataset, as explained in detail in the Methods. Briefly, starting from 469 candidate proteins meeting initial selection criteria (minimum length of 500 aa, functional annotation as tail fibre or tail spike, predicted presence of a parallel $\beta$-helix domain), we clustered these into 124 sequence groups, modelled all representatives using AlphaFold2 [57], and manually evaluated the resulting structures for conserved features characteristic of depolymerases, particularly the parallel $\beta$-helix fold. This process yielded 50 candidate sequences from 32 clusters for recombinant protein expression (see Methods). For each of these 50 candidates, we attempted recombinant production and purification, followed by enzymatic testing against diverse *Klebsiella* strains representing 118 K-types (S2 Table). Out of the 50 candidates, 34 recombinant proteins received insufficient yield, 2 were produced as soluble proteins but showed no detectable activity on any of the 118 K-types, and 14 yielded soluble, active enzymes targeting one of 11 K-types (KL23, KL28, KL32, KL38, KL46, KL52, KL60, KL62, KL64, KL127 and KL143; see Fig 4A, S4A Table and S2 Text).

Given that 34 of the 50 recombinant expression attempts resulted in low or undetectable yields and no measurable activity, we next examined whether insufficient expression could account for the apparent lack of function. Notably, several of the depolymerases that we confirmed as active were also expressed at low levels, yet even small amounts of protein were sufficient to degrade capsule material (see S4A Table). These conditions were consistent with, and in some cases directly matched, those reported in the literature for successful expression of large *Klebsiella* phage depolymerases (see S4B Table). To assess whether expression constraints were responsible for the inactivity of the remaining candidates, we undertook a systematic re-expression campaign in which eight cluster representative proteins were expressed under a new set of conditions (IPTG inducer concentration, expression temperature, density of the culture induced, and expression duration). Despite these modifications, seven of the eight candidates showed no detectable overexpression by SDS–PAGE, and the single protein that did express at low levels remained inactive against all tested *Klebsiella* isolates (S4C Table).

Structural modelling of non-expressed proteins revealed no obvious folding defects, with predicted architectures indistinguishable from active homologues (S5 and S6 Figs). Moreover, there was no evidence that prophages from which low-yield proteins originated were more likely to be cryptic (S7 Fig) and, in several cases, co-encoded both active and inactive depolymerases (S8 Fig). We also found two pairs of highly similar proteins (amino acid identity > 97%, but modification accumulation in different enzyme domains), where one was recombinantly obtained as soluble and active, and the other was not expressed (S9 Fig). These findings suggest that the failure to obtain soluble, active enzymes is unlikely to be attributable to cryptic prophages or obvious structural degradation, but rather reflects the difficulty of heterologously expressing large, trimeric proteins outside their native phage context.

The 26 GWAS predictors (from both the 'strong' and 'likely' categories; see also S3 Table) were first compared to the manually curated collection of 50 candidate depolymerases from the KASPAH-REF isolates (S4A Table) based on their protein sequence similarity. When the GWAS predictor from the 'strong' category did not not display sequence similarity to any protein within our recombinant-protein set, or any of the available enzymes with confirmed specificity described already in the literature (S1 Table), we commissioned synthetic plasmid preparation and recombinant protein production for one representative protein from each cluster based on manual analyses to maximise chances of its successful production and enzymatic activity (see Methods).

Fig 4 shows a direct comparison of computational and experimental results per K-locus, alongside additional recombinant proteins from the literature from virulent phages. Specifically, Fig 4A shows a table of the number of predictors for

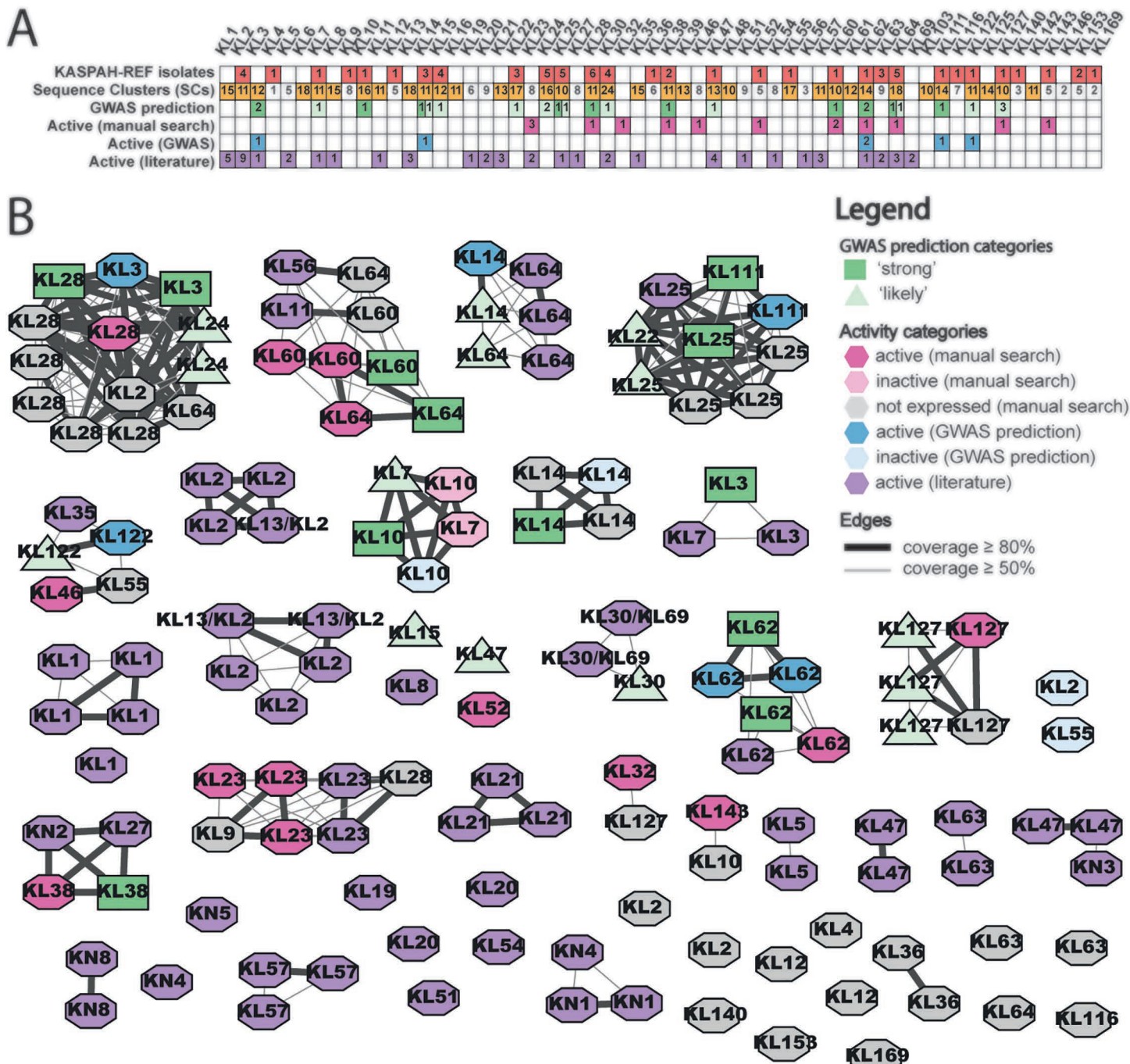

**Fig 4. Prophages can be an unpredictable source of active depolymerases. (A)** Table shows a systematic comparison of the number of bacterial SCs from the KASPAH-REF collection (red), the number of bacterial SCs from the full dataset (yellow), number of GWAS predictions in the 'strong' (dark green) 'likely' (light green) category, and the number of active recombinant proteins based on manual (magenta) and based on GWAS' 'strong' predictors (dark blue), and the number of experimentally verified depolymerases from virulent phages from Cheetham and colleagues [27]. **(B)** Sequence similarity network where each node represents a protein sequence and edges link proteins with a minimum of 50% bidirectional coverage (thin edges) or a minimum of 80% bidirectional coverage (thick edges). Colours are in accordance with panel A, with blue nodes showing additional proteins obtained via gene synthesis and recombinant protein production. The data underlying this Figure can be found at Figshare (https://doi.org/10.6084/m9.figshare.29181188), S1 Table, S3 Table, S4 Table, S1 Data, and can be reproduced using code archived in Zenodo (https://doi.org/10.5281/zenodo.18699826).

each K-locus, and Fig 4B shows a network-like comparison of the two sets of proteins, alongside 58 previously published depolymerases from virulent phages [27] (see S1 Table). We found that, out of 12 predictors in the 'strong' category, 5 were confirmed by sequence or structural similarity to the recombinant protein analysis (KL28, KL38, KL60, KL62, KL64) and 2 were confirmed by sequence or structural similarity to recombinant proteins from the literature: KL3 had distant similarity and a visibly different C-terminus, while the KL25 prediction had structurally identical receptor-binding domain (see S3 Text). Of the remaining 5 predictors, which were used to select 5 proteins for commercial production, 3 were confirmed as active enzymes (KL3, KL62 and KL111) while 2 did yield soluble proteins (KL10 and KL14) but no activity was detected against any of the 118 K-types. Additionally, one protein commercially produced based on a 'likely' predictor (KL122) was found active against isolates with the KL122 locus (Fig 4B; S4A Table).

By contrast, out of 11 K-types for which we obtained active enzymes via the manual search in the KASPAH-REF collection, five specificities (K23, K32, K46, K52, KL143) could not be compared with GWAS results, as these K-types were excluded from the set of 35 loci, hence no predictors were generated. (For one of them, KL23, a virulent-phage protein had previously been reported in the literature and we observed structural homology of the receptor-binding domain; see S3 Text.) The enzymatic activity of the remaining six KASPAH-REF K-specific enzymes (K28, K38, K60, K62, K64 and KL127) was in line with GWAS predictions, either predictors classified as 'strong' or 'likely'. Importantly, 5 of 14 enzymes were active against a different K-type than the K-type of the prophage host (see S4 Fig).

In summary, we found that (a) of the 12 'strong' GWAS predictors, 10 were experimentally validated and 2 remained inconclusive, and (b) among the 14 active enzymes identified through manual search, only 6 corresponded to the 35 K-loci included in the GWAS analysis; all 6 were supported by GWAS predictions ('strong' or 'likely' predictors). Importantly, prophages proved to be an unpredictable source of active enzymes as most candidate proteins resulted in a low yield, and several enzymes displayed activity against a different K-type than that of their host strain (see also Discussion).

## Complexity of the genotype–phenotype relationships in phage depolymerases

While the overall agreement between GWAS predictions and experimental validation was strong, closer inspection of the sequence similarity network (Fig 4B) revealed multiple genetic links between enzymes with different K-type specificities (confirmed or predicted). To explore this complexity more systematically, we performed a detailed comparison of the protein sequence, structure and specificity relationships across these enzymes, as described in S3 Text. This analysis revealed several key patterns that exemplify the challenges in predicting substrate specificity of these enzymes from sequence – or even structure – alone.

First, we observed frequent modularity, particularly variation in N-terminal domains known to be related to the virion taxonomic or morphological groups and to vary among phage depolymerases [49] (see S3 Text). Second, we found that enzymes that shared sequence and structural similarity often exhibited distinct substrate specificities, indicating that proteins with related structures of their receptor-binding domains might be functionally divergent. Fig 5 demonstrates this using the example of enzymes known or predicted to target K-types KL3, KL24 and KL28 from two connected components of the network in Fig 4B. We found that GWAS predictors for KL3 (PC0449) and KL28 (PC0279) were both confirmed experimentally (391_03 and 1251_37); a comparison revealed that they are genetically and structurally related to each other and to the KL24 predictions (PC0406 and PC1397; Fig 5A). Nevertheless, recombinant depolymerase against K3 (391_03) and against K28 (1251_37) did not cross-react against K3/K24/K28 reference strains (Fig 5B) even though the K3, K24 and K28 polysaccharide structures are biochemically related (Fig 5C). Furthermore, comparison to another recombinantly tested depolymerase (NCBI accession YP_03347555.1) from a virulent phage active on a K3-type capsule showed no genetic or structural similarity to any of the other ones.

The network analysis (Fig 4B) further revealed depolymerases with identical substrate specificity whose structural models indicate homology, even though their primary sequences are markedly divergent. Examples include the K1-specific

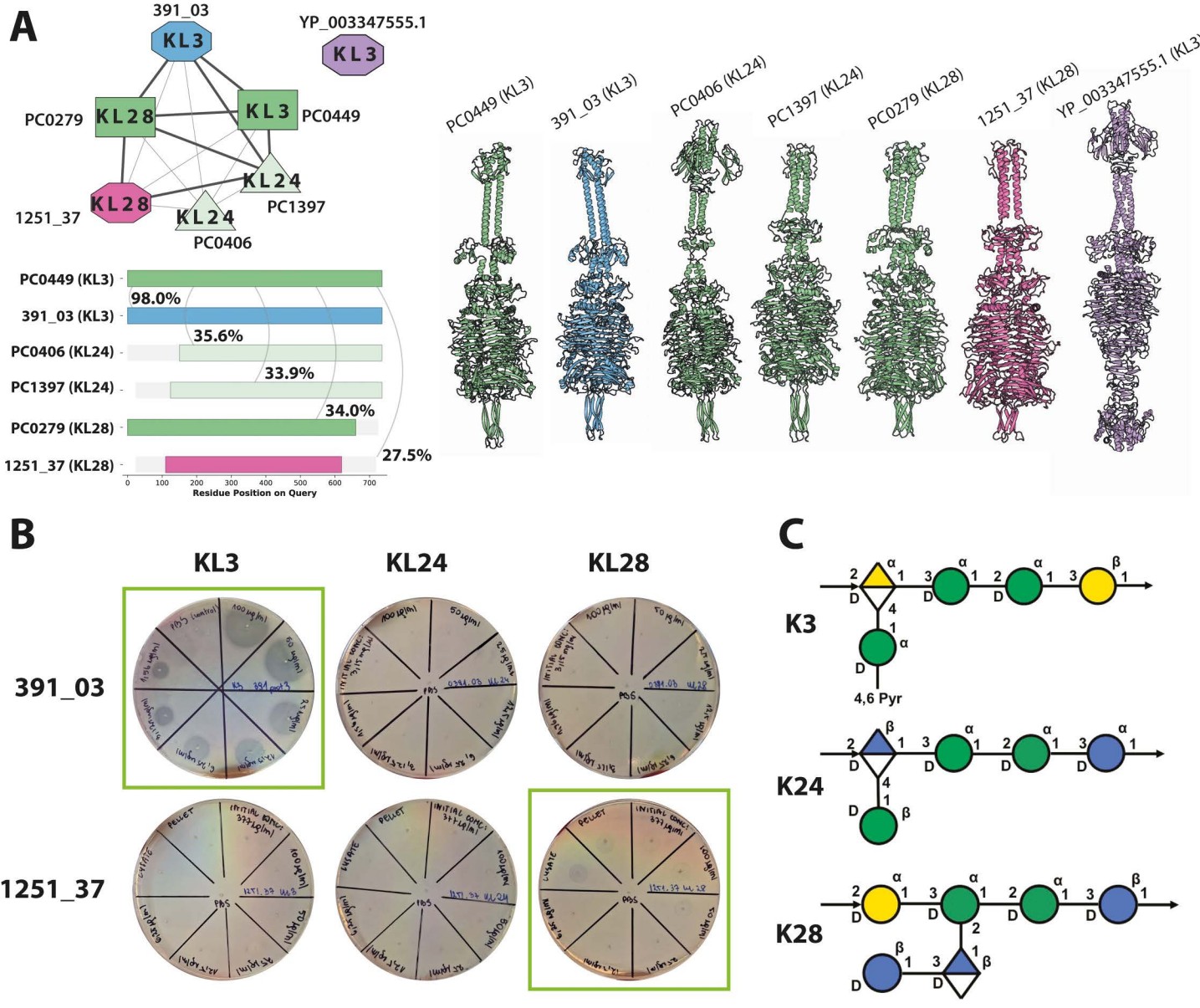

**Fig 5. Depolymerase specificity is a complex phenotype. (A)** One of the connected components from Fig 4B showing seven depolymerases: two 'strong' GWAS predictions (KL3, KL28; dark green), two 'likely' GWAS predictions (KL24; light green), one active protein from the manual search (KL28; pink), one active protein from GWAS prediction (KL3; blue), and an active protein from a virulent phage (KL3; purple). All of these proteins, except the virulent KL3 depolymerase, are genetically and structurally related, seen in a systematic visualisation of pairwise alignments and 3D models obtained via AlphaFold3. **(B)** Spot test assay validating the predicted capsular specificity of 391_03 and 1251_37 selected proteins against reference K-loci KL3, KL24 and KL28. **(C)** Schematic representations of the K3, K24, and K28 capsule polysaccharides, as obtained from K-PAM [16]. The data underlying this Figure can be found in S1 Table, S4 Table, and can be reproduced using code archived in Zenodo (https://doi.org/10.5281/zenodo.18699826).

depolymerases (NCBI accessions BAW85698.1 and YP_009302756.1, protein sequence identity of 35%), K5-specific depolymerases (APZ82848.1 and WNO47121.1, identity of 32%) or K21-specific depolymerases (BAW85693.1 and YP_003347556.1, identity of 41%). Therefore, our results indicate that genetic or structural similarity is often insufficient to infer depolymerase specificity, pointing to a nuanced genotype–phenotype map of these enzymes.

### *K. pneumoniae* prophage RBPs contain diverse enzymes

Our results showed that the $\beta$-helix-containing depolymerases are present in prophages, are relatively rare, highly diverse and not always active. Additionally, the 'pectin lyase-like' domain was not the predominant domain detected amongst the top GWAS results. Therefore, we next investigated the prevalence of different protein domains in prophage RBPs from our dataset. To this end, we analysed ECOD domain diversity across 4,598 high-completeness prophages, focusing on prophage protein clusters annotated as RBPs. For each cluster, we classified detected ECOD domains and quantified K-locus diversity via the Simpson index, which reflects the variation of K-loci associated with bacteria in which these proteins were found. We reasoned that capsule-specific RBPs should predominantly occur in genomes sharing the same K-locus.

The association between ECOD domains and K-locus diversity varied substantially across functions. Only four ECOD domains were linked to protein clusters predominantly associated with a single K-locus or few K-loci: *Phage tail fiber protein trimerization domain* (T-level: 79.1.1), *Intramolecular chaperone domain in virus tail spike protein* (T-level: 3240.1.1), *Pectin lyase-like* (T-level: 207.2.1), and *SGNH hydrolase* (T-level: 2007.5.1; see Fig 6A). Among these, only the *Phage tail fiber protein trimerization domain* and the *Pectin lyase-like* domain were predominantly found in clusters with low K-locus diversity, suggesting a higher degree of K-locus association. In contrast, the *Intramolecular chaperone domain in virus tail spike protein* (T-level: 3240.1.1), *gp11/gp12 receptor-binding domain* (T-level: 877.1.1), and *Uncharacterized protein CV0426* (T-level: 3809.1.1) tended to occur in clusters spanning multiple K-loci, indicating broader distribution. Notably, the *SGNH hydrolase* domain exhibited the greatest variability in K-locus diversity, and on average, was less K-locus-specific than the *Pectin lyase-like* domain.

To investigate how the frequency of RBP domains varies between different K-loci, we mapped the ECOD domains from Fig 6A on the high-quality prophages found in bacterial genomes with various K-loci (see Fig 6B). Most of the RBPs detected carried the 'SGNH hydrolase' domain, while the 'Pectin lyase-like' domain was on average less prevalent, usually co-occurring with other domains (see S10 Fig). Furthermore, we found examples where while no domains were detected in prophage RBPs, adjacent proteins contained hits to 'SGNH hydrolase' (see S11 Fig). Altogether, these results highlight the structural diversity of RBPs found in *K. pneumoniae* prophages and point to the prevalence and importance of alternative capsule recognition mechanisms like SGNH hydrolases in interactions between bacteria and their temperate phages.

### Tail fibres with SGNH hydrolase domains are functional esterases

Our analyses revealed that SGNH hydrolases represent one of the most frequent domain families in prophage receptor-binding proteins (RBPs) of *K. pneumoniae*. SGNH-containing RBPs appeared as recurrent GWAS hits across multiple K-loci and, in some K-loci like KL2, KL6 or KL11, nearly all prophages carried a RBP with this domain. A RBP with an SGNH hydrolase domain was previously shown as an active esterase in an *E. coli* phage G7C, which acts as a deacetylase removing an O3-acetyl group from the bacterial lipopolysaccharide [58]. Together, the prevalence of SGNH-domain RBPs across multiple K-loci and their repeated identification as GWAS predictors led us to hypothesise that these prophage-encoded proteins function as active acetyl esterases.

To assess the enzymatic activity of SGNH-domain RBPs in *Klebsiella* prophages, we recombinantly produced two proteins encoded by the strongest GWAS-predicted SGNH-domain candidates: protein 164_08, selected based on the GWAS predictor for KL2 (hereafter 164_08-KL2), and protein 174_38, selected based on the GWAS predictor for KL55 (hereafter 174_38-KL55; cf. Fig 3). AlphaFold3 modelling showed that both proteins are tail fibres with a characteristic SGNH-containing, brush-like C-terminal architecture (Fig 7A). As a negative control, we included 184_43, selected based on the GWAS predictor for KL111 (hereafter 184_43-KL111), which had a parallel $\beta$-helix domain but lacked an SGNH domain (see Fig 7A). To functionally characterise these proteins, we employed an established surrogate assay based on hydrolysis of *p*-nitrophenyl acetate (pNPA). SGNH hydrolases – including characterised carbohydrate esterases and

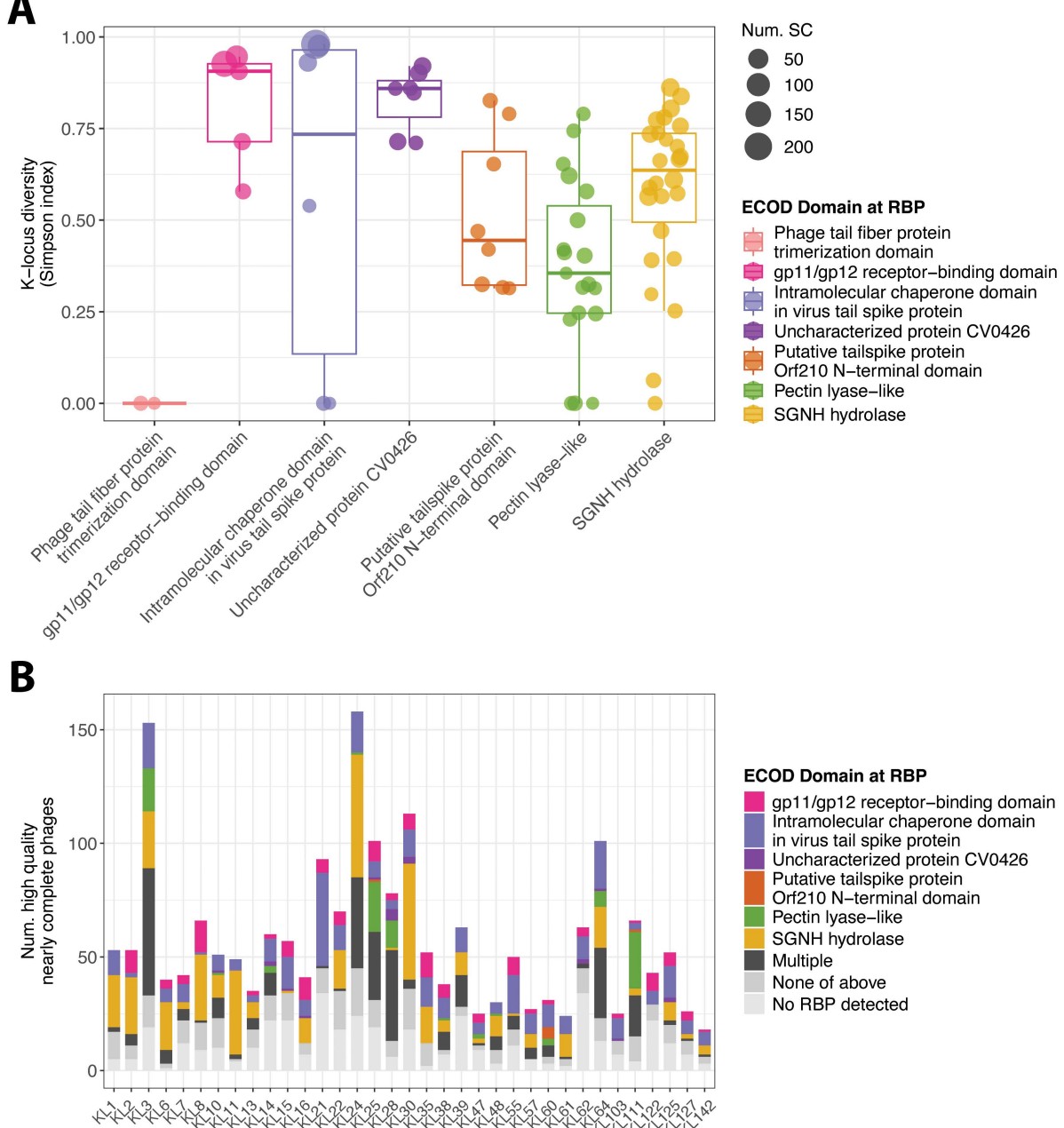

**Fig 6. SGNH hydrolases are prevalent amongst *K. pneumoniae* prophage RBPs. (A)** Capsule-type (K-locus) diversity within RBP clusters, coloured by their dominant ECOD domain. Prophage proteins from four *Klebsiella* species were clustered at ≥50% identity and ≥80% coverage, then RBPs were identified by PHROG annotation (tail fiber or tail spike; probability ≥90%, qcov ≥ 50%, scov ≥ 50%). ECOD domains were assigned to each RBP cluster (probability ≥ 90%, tcov ≥ 10%), allowing multiple domains per cluster. Each point shows a cluster with ≥10 members from ≥3 SCs, its Simpson index of K-locus diversity (0 = uniform, 1 = maximally diverse), and its ECOD domain. **(B)** Number of high-quality, nearly complete prophages encoding each ECOD domain within PHROG-defined RBPs, stratified by K-locus (completeness of ≥90%, *n* = 4,598, of which *n* = 2,047 belonged to the 35 most diverse K-loci). Bars are coloured by the single ECOD domain detected (or light grey if no PHROG RBP was detected, dark grey if multiple ECOD domains co-occur). The most frequent multi-domain combinations include gp11/gp12 + SGNH hydrolase and pectin-lyase-like + Orf210 N-terminal or gp11/gp12 domains (see S10 Fig). Prophages whose RBPs lack any of the domains shown in (A) often encode adjacent proteins with other enzymatic folds (e.g., cysteine proteases, triphosphate hydrolases, SGNH hydrolases; S11 Fig). The data underlying this Figure can be found at Figshare (https://doi.org/10.6084/m9.figshare.29181188), and can be reproduced using code archived in Zenodo (https://doi.org/10.5281/zenodo.18699826).

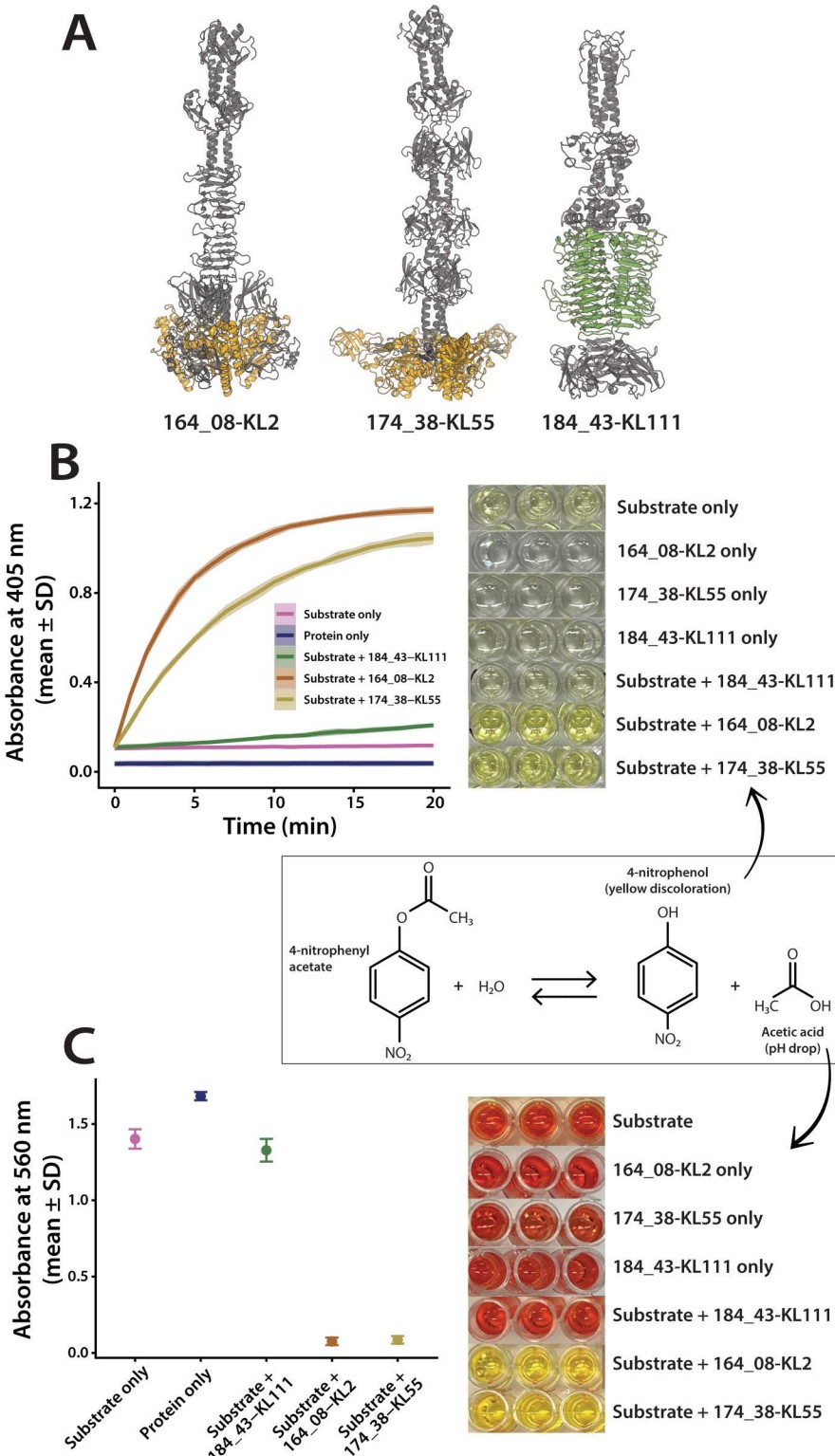

**Fig 7. Esterase activity of SGNH-containing tail fibres against 4-nitrophenyl acetate. (A)** AlphaFold3 models of three recombinantly produced proteins: 164_08-KL2 (GWAS prediction for KL2), 174_38-KL55 (GWAS prediction for KL55) and 184_43-KL111 (GWAS prediction against KL111). Both 164_08-KL2 and 174_38-KL55 are tail fibres with an SGNH hydrolase domain (yellow) and did not show depolymerase activity against any of the 118

reference K-types; 184_43-KL111 was a depolymerase with an expected parallel $\beta$-helix domain (green) and was active against the KL111 strain. **(B)** Time-course of absorbance at 405 nm showing formation of the yellow product 4-nitrophenol in reactions containing substrate with 164_08-KL2, 174_38-KL55 or 184_43-KL111. The "Protein only" line represents the mean of protein-only controls with no substrate measured independently for 164_08-KL2, 174_38-KL55 and 184_43-KL111. As these control traces were time-independent and overlapped within variability, a single mean line is shown for clarity. The panel includes photographs of the corresponding reaction wells after 20 min, showing yellow coloration in the 164_08-KL2 and 174_38-KL55 reactions, mild yellow coloration in the substrate-alone wells consistent with substrate autohydrolysis, and a schematic representation of the hydrolysis reaction generating $p$-nitrophenol and acetic acid (black box). The shaded area marks one standard deviation from the mean. **(C)** Absorbance at 560 nm measured after the addition of phenol red, comparing substrate alone, the mean of "Protein only" controls (164_08-KL2, 174_38-KL55, and 184_43-KL111), and reactions containing substrate with 184_43-KL111, 164_08-KL2, or 174_38-KL55. Error bars represent one standard deviation from the mean. The panel includes photographs of the wells showing red coloration in controls and 184_43-KL111, and yellow coloration in the 164_08-KL2 and 174_38-KL55 reactions. The data underlying this Figure can be found in S4 Table, S3 Data, and can be reproduced using code archived in Zenodo (https://doi.org/10.5281/zenodo.18699826).

polysaccharide deacetylases – readily cleave simple acetate esters such as pNPA, and this reaction is widely used as a diagnostic readout of SGNH catalytic activity independent of the native polysaccharide substrate [59–61]. Hydrolysis of pNPA releases $p$-nitrophenol and acetic acid, enabling robust detection through absorbance at 405 nm and pH-dependent colourimetric shifts, respectively.

Both 164_08-KL2 and 174_38-KL55 efficiently hydrolysed pNPA, exhibiting a clear time-dependent increase in absorbance and visible yellowing consistent with $p$-nitrophenol formation (Fig 7B). A complementary phenol-red pH-shift assay confirmed acid release during the reaction, further supporting ester cleavage (Fig 7C). In contrast, 184_43-KL111 and all no-substrate controls showed no detectable activity. Together, these assays demonstrate that the SGNH-domain tail fibres encoded by *Klebsiella* prophages function as active acetyl esterases, supporting their proposed role in capsule modification rather than depolymerisation (see Discussion).

## Discussion

In this work, we aimed to uncover genetic and molecular determinants of capsular specificity in temperate phages of *K. pneumoniae*. Specifically, we aimed to help fill three major knowledge gap areas in the context of *Klebsiella*-phage interactions, namely (i) which viral genes determine capsular specificity, (ii) why depolymerases are difficult to find in prophages and (iii) whether predicted capsule-specific depolymerases are functionally active. To this end, we carried out a genome-wide association study (GWAS) on an ecologically diverse dataset of 3,900 bacterial genomes, looking for genetic predictors from their 8,105 prophage regions that correlate with the K-type predictions (K-loci); independently, we carried out a manual search by recombinantly producing over 60 putative depolymerases and testing against a broad panel of *Klebsiella* K-types. The computational approach provides important insight into the first two of these questions (capsular specificity versus depolymerases), while the comparison of the computational and experimental parts provides insight into the third question (functional activity).

A major advantage of our GWAS-based approach over many machine-learning methods is that it does not rely on any predefined protein functions. This function-agnostic design allowed us to explore which prophage genes determine capsular specificity and why identifiable depolymerases are so rare. In around 30% of K-loci, the analysis yielded no significant predictors under any threshold, suggesting that no single protein family perfectly predicts the distribution of any K-locus. Even for the K-loci with the strongest statistical signal, none achieved near-perfect classification ($F_1 \approx 1$), with only four exceeding $F_1$ of 0.8 and most remaining below 0.5. These results mirror those of a complementary study by Concha-Eloko and colleagues [34], whose larger deep-learning framework correctly matched 48/164 depolymerases ($\sim 29\%$) to the correct K-locus using the top-ranked prediction, underscoring that depolymerase sequence alone is often a weak predictor of capsular specificity when inferred from lysogenic genomic data. Among our strongest predictors across all K-loci, we observed a striking diversity of prophage functions, dominated by tailspikes and tail fibres carrying either the classical $\beta$-helix depolymerase domain or the SGNH hydrolase domain, alongside previously unrecognised receptor-binding

proteins and even putative acetyltransferases. A systematic manual search further revealed that many prophages lacked identifiable RBPs altogether–possibly reflecting rapid sequence diversification or incomplete assembly of tail-fibre genes in short-read genomes. The case of the KL1 locus illustrates this complexity well: prophages in KL1 strains encoded a mosaic of SGNH-domain RBPs, RBP clusters shared across distinct K-loci, and in one case no RBP gene at all (see S12 Fig). Collectively, these findings indicate that capsular specificity in temperate *Klebsiella* phages is a complex phenotype. Classical polysaccharide depolymerases represent a major mechanism of host recognition, consistent with the prevailing model derived from virulent phages, but additional enzymatic strategies–including SGNH-domain esterases–are widespread and likely contribute to capsule interaction in a modification-dependent manner.

A large set of recombinantly produced depolymerases revealed that only 14 out of 50 proteins from the manual search showed evidence of capsular specificity against one of 118 reference K-types. Of those 14, five were active on a different K-type than that predicted of their host. The comparison of the computational and experimental approach revealed that, where GWAS and manual datasets overlapped, predictions and experimental results were largely concordant, except for two cases: one lacking the C-terminus and another that was expressed but inactive. Importantly, however, of the remaining 36 recombinant proteins, 34 were not overexpressed. Our attempts to optimise expression conditions for 8 representative of those 34 proteins resulted in an improved expression of only one protein, but it remained inactive. We did not find support for the hypothesis that these proteins originate from cryptic prophages because their genomes were complete with no excess of transposons, and many were recently integrated, suggesting they are likely infectious. We also found co-occurrence of active and non-expressed depolymerases in the same prophage genome in three cases. Comparison of predicted structures suggested that the non-expressed depolymerases are not defective, and we identified two pairs of enzymes with near-identical sequences in which one protein was successfully expressed and enzymatically active, while the other failed to overexpress. Taken together, these results indicate that recombinant expression and enzymatic activity of prophage depolymerases are highly sensitive to both sequence and construct context. Consequently, functional activity cannot be reliably inferred from sequence identity or structural prediction alone.

The relatively high frequency of the 'SGNH hydrolase' domain among GWAS predictions underscores its importance as a host-range determinant in temperate phages of *K. pneumoniae*. This prevalence is also reflected by the high proportion of RBPs with this domain across all prophages in our dataset (cf., Fig 6). The SGNH hydrolase domain, encompassing a diverse family of esterases and lipases, has previously been identified as a structural topology in some RBPs of other phages [62], including those infecting *Acinetobacter baumannii* [63] and *E. coli* [58]. Furthermore, Prokhorov and colleagues demonstrated that gp63.1, a SGNH-containing RBP from an *E. coli* phage G7C, was an esterase functioning as a deacetylase, removing an O3-acetyl group from lipopolysaccharide without degrading the O-chain [64] – a reaction essential for G7C's infectivity. To see if the SGNH hydrolases observed in this study might have a similar function, we produced two proteins, classified as strong predictions by GWAS: 164_08 (KL2 prediction) and 174_38 (KL55 prediction). We then confirmed esterase activity for both proteins using a colorimetric assay with p-nitrophenyl acetate, which detects hydrolysis by the release of p-nitrophenol and acetic acid, consistent with an active SGNH-type esterase fold capable of cleaving acetyl esters. These results suggest that deacetylation may represent a common alternative strategy in temperate *Klebsiella* phages to recognise and "shave" polysaccharides from acetyl groups (without depolymerisation) to initiate infection.

The idea that temperate phages of *K. pneumoniae* may often rely on deacetylation to initiate infection is consistent with the fact that O-acetylation of bacterial polysaccharides is common, with multiple lines of evidence. O-acetylation has been detected in many K-types, either as part of the defined repeating-unit structure (e.g., K5, K21b, K22, K33, K54, K55, K58, K59, K82) [65–73] or observed in NMR spectra without precise localisation (e.g., K4, K8, K10, K11, K13, K20, K24, K30, K41, K43, K44, K49, K63, K64, K69) [74–88]. Likewise, K-loci in *K. pneumoniae* often feature genes annotated as acetyltransferases [18]. Incidentally, we also detected putative acetyltransferases in two of our strongest GWAS hits–despite filtering out short genes–and between 5%–30% of prophages per K-locus in our dataset encode proteins with a predicted acetyltransferase function (see S13 Fig), suggesting that sugar modification may be driven – at least partially – by

incoming prophages. As the presence or absence of an acetyl group in otherwise identical or nearly-identical polysaccharide structures can be associated with different serological assignment, as in the case of K22/K37 [67,89] or K30/K33 [68,81], it is plausible that current serotyping and K-locus genotyping underestimate capsule diversity, as O-acetylation introduces biochemical variation that can affect phage recognition, host tropism and immune interactions beyond what is captured by these schemes. However, further work will be required to determine how capsule acetylation state influences adsorption efficiency or infection outcomes in vitro and in vivo.

If the dynamic interplay between acetylation and deacetylation shapes host tropism of bacterial viruses, this may reflect both a direct mechanism of host recognition by temperate phages and a competitive strategy to modify host surfaces in ways that interfere with other phages, particularly those that rely on unmodified polysaccharides. Similar mechanisms have been reported across gram-negative bacteria, where O-acetylation of surface polysaccharides alters phage susceptibility: in *E. coli* and *Salmonella enterica* serovar Typhimurium, it modulates sensitivity to superinfecting phages [90,91]; in *Shigella flexneri*, temperate phage-mediated O-acetylation drives serotype conversion and can confer phage resistance [92,93]; and in *Pseudomonas aeruginosa*, phage D3 introduces O-acetyl modifications to prevent superinfection by related phages [94,95]. While in *K. pneumoniae* some depolymerases may tolerate acetylated substrates [96], others may not, suggesting that these subtle modifications could significantly reshape the phage infectivity landscape. Overall, these findings highlight polysaccharide acetylation and deacetylation as a potentially underappreciated layer of host-virus specificity and phage–phage competition (e.g., via superinfection exclusion mechanisms) in *K. pneumoniae*.

A key limitation of our study is that the dataset underlying the GWAS was derived from isolates collected in only two countries–Australia and Italy–which inevitably constrains the total genetic diversity represented. However, this focused sampling also minimises biases that often affect public databases, which are typically enriched for clinical and antibiotic-resistant isolates. Our collection was assembled independently of resistance or virulence traits and thus provides a less epidemiologically biased view of prophage diversity. To assess how representative our dataset is of the broader species diversity, we compared its sequence type (ST) distribution to two global *K. pneumoniae* resources: ARMnet, which comprises over 45,000 publicly available genomes with standardised epidemiological typing from Pathogenwatch [97], and KlebNNSsero [98], a geographically broad set of around 2,000 neonatal bloodstream isolates assembled without selection for resistance or virulence (see S14 Fig). We found that 390 of 542 STs (72%) have been detected in at least two countries, and 347 STs (64%) have been reported on at least two continents, including globally distributed lineages such as ST11, ST15 and ST307. Notably, 382 STs (70%) were detected in at least one additional country beyond Italy and Australia, and 311 STs (57%) were detected on at least one additional continent beyond Europe and Oceania. Altogether, this suggests that our dataset captures a representative cross-section of the species' global diversity while maintaining statistical tractability and biological consistency for GWAS analyses.

Beyond sampling, additional limitations concern the analytical and experimental scope of the study. Our GWAS was restricted to genomes with confidently assigned K-loci, which supports biological relevance but limits statistical power by excluding rarer K-types and other species from the *Klebsiella* genus where the K-typing does not work as well. Moreover, GWAS results are inherently sensitive to parameter choices–including the definition of protein clusters, K-locus prediction accuracy, and population-structure correction–and future work combining larger datasets with tuned parameters will likely recover additional associations. We also considered whether degraded or cryptic prophages might bias GWAS results, as non-functional prophages could accumulate mutations unrelated to capsular specificity. However, when we repeated the analyses using only high-quality prophages (CheckV completeness $\geqslant 99\%$), the results were strongly correlated with those from the full dataset ($r = 0.61 – 0.84$; S15 Fig), indicating that degraded prophages did not significantly influence the associations. Another key limitation is that our experimental validation focused on depolymerase activity detectable via halo formation. As a result, we may have missed RBPs with enzymatic activity that do not produce visible halos or operate via alternative interaction mechanisms. Future work could address this using complementary biochemical and structural approaches, such as quantitative polysaccharide degradation assays or mass spectrometry–based analyses, although

these methods are inherently lower throughput and require purified capsule substrates. Finally, while our findings highlight unexpected complexity in capsule recognition among temperate *Klebsiella* phages, it remains unclear to what extent these insights generalise beyond this species or to other phage lifestyles such as strictly lytic (virulent) phages.

## Conclusions

Our combined GWAS and experimental screen in a near-random sampling of *K. pneumoniae* prophages demonstrates that, although classical right-handed $\beta$-helix depolymerases do indeed mediate capsular specificity in temperate phages, they are far from the sole mechanism at play. SGNH hydrolase–driven deacetylation emerges as a likely and functionally dominant strategy for capsule recognition and modifying the capsule, revealing a previously unappreciated layer of molecular complexity. Together, our findings reveal an unexpectedly diverse repertoire of prophage-encoded capsule-interacting enzymes. This diversity suggests that capsule structure and modification state may influence phage susceptibility in more complex ways than previously appreciated. A deeper understanding of these mechanisms will be important for future efforts aiming to harness phages therapeutically or to design capsule-based interventions.

## Materials and methods

### Computational approach

**Data.** To build a representative genome dataset of *Klebsiella spp.*, we combined two collections. The first collection (KASPAH) consisted of 391 *Klebsiella pneumoniae* species complex isolates (*K. pneumoniae*, *K. quasipneumoniae*, *K. variicola*), obtained from a diagnostic laboratory serving four major hospitals in Melbourne between April 2013 and March 2014 [39,41,42]. These isolates were sourced from various clinical samples including pneumonia ($n = 44$), rectal swabs ($n = 82$), throat swabs ($n = 14$), urinary tract infections (UTI; $n = 193$), wound or tissue ($n = 28$), disseminated infections ($n = 25$), and other sources ($n = 5$). Sequencing was performed using short-read Illumina technology for all isolates, while 99 were additionally sequenced using long-read Oxford Nanopore Technology (ONT) to generate hybrid short/long-read finished assemblies [43]. The second collection (KLEBPAVIA) encompasses 3,510 *Klebsiella spp.* isolates chosen from 6,548 samples from diagnostic laboratories in Pavia, Italy, and its surroundings between June 2017 and November 2018 [40]. This dataset includes 3,482 *Klebsiella spp.* isolates, with an additional 27 *K. quasipneumoniae* isolates not published in Thorpe and colleagues 2022, as well as a single outgroup isolate originally classified as *Klebsiella* but later identified as *Superficieibacter maynardsmithii* [99]. The full dataset consists of 3,900 isolates of *Klebsiella spp.* and a single outgroup. Importantly, the combined collection consists of diverse *Klebsiella* isolates from multiple ecological niches, hence providing a diverse overview of both capsule and prophage diversity within. All genomes were annotated with the PATRIC server between November 2022 and March 2023 and with the recipe Bacteria / Archaea and Taxonomy ID 573 (*K. pneumoniae*). The list of bacterial genomes is provided in S1 Data.

**Inference of the population structure.** Core genes were defined on the whole dataset (including the outgroup) using panaroo v1.3.4 [100] on 'strict' mode, as genes present in at least 99% of isolates. SNPs were extracted from an alignment of 1,668 core genes with snp-sites v2.5.1 [101]. The maximum likelihood phylogenetic tree was inferred from those SNPs using IQ-TREE v2.1.4 [102] with the following parameters: `-t PARS -ninit 2` and the number of constant sites obtained from snp-sites `-C`. The best-fitting model of sequence evolution was identified by IQTREE ModelFinder as GTR+F+I+R10. Independently, we ran Kleborate v2.4.1 [103] to predict *Klebsiella* species for all 3,900 isolates, yielding 19 different species: *K. pneumoniae*, *K. variicola*, *K. quasipneumoniae subsp. quasipneumoniae*, *K. quasipneumoniae subsp. similipneumoniae*, *K. oxytoca*, *K. michiganensis*, *K. ornithinolytica*, *K. aerogenes*, *K. pasteurii*, *K. grimontii*, *K. planticola*, *K. quasivariicola*, *K. spallanzanii*, *K. huaxiensis*, *K. Ka3*, *K. terrigena*, *K. Ko10*, *K. huaxiensisNEW* and *K. Kterr4*. We then used PopPunk v2.6.3 to identify bacterial genome sequence clusters (SCs) within each species, which were used as a proxy for bacterial lineages. Four of these species (*K. Ka3*, *K. Ko10*, *K. huaxiensisNEW*, *K. Kterr4*) are outgroups of other species (respectively for *K. aerogenes*, *K. huaxiensis*, *K. huaxiensis* and *K. quasiterrigena*) and

were introducing noise in the clustering, hence were not considered for this analysis. The remaining 3,876 genomes assigned to 15 species were analysed with the same approach as described by Thorpe and colleagues [40]. Briefly, for each species we fit the mixture model of PopPunk v2.6.3 [44]. The number of components (*k*) was chosen based on the scatter plot of core and accessory distances and iteratively improved. We then ran the model refinement module of PopPunk for each species individually. However, in the case of two species (*K. spallanzanii* and *K. quasivariicola*) the refinement introduced noise in the clustering and hence was abandoned. We compared the resulting clusters to those from Thorpe and colleagues with the Adjusted Rand Index (ARI) using python `sklearn.metrics.adjusted_rand_score`, for which a value of 1 indicates perfect agreement between the two clusterings and 0 indicates random labelling. We compared the resulting trees with microreact for species with ARI < 0.8, all of them appearing coherent despite the discrepancies. The resulting PopPunk clusters are henceforth referred to as bacterial sequence clusters (SCs).

**K-locus identification.** We used Kaptive v2.0 [18], implemented in Kleborate v2.4.1 [103], to predict capsule polysaccharide synthesis loci (K-loci) for all isolates, which are predictive of capsular types (K-types). Species and K-loci were predicted with Kleborate for all isolates. To identify *Klebsiella* species for which K-locus prediction is reliable, we searched for the species in which 80% of isolates or more had a confident K-locus prediction (i.e., assigned 'Good', 'Very Good' or 'Perfect' by Kaptive) and which had more than 5 members. The resulting dataset consisted of 2,527 genomes of isolates belonging to four species from the *K. pneumoniae* species complex (*K. pneumoniae, K. variicola subsp. variicola, K. quasipneumoniae subsp. quasipneumoniae, K. quasipneumoniae subsp. similipneumoniae*), henceforth referred to as the GWAS dataset.

**Identification of prophages.** To detect prophages in bacterial genomes, we used two complementary tools: VirSorter v1.0.5 [45] and PhiSpy v4.2.6 [46]. VirSorter predicts clusters of viral genes based on sequence similarity to its database (`-db 1`), though it frequently over-predicts the length of prophage regions. PhiSpy, which combines heuristic methods, including protein length, transcription strand directionality or AT/GC skew, tends to predict shorter prophage regions and may capture other mobile genetic elements. As the goal was to minimise the false-negative rate of prophage prediction to consider as many likely viral proteins in the GWAS approach, we took the union of the prophage regions predicted by both tools, creating a primary detection. Manual investigation of the resulting putative viral regions indicated an occasional presence of poorly annotated receptor-binding proteins. Hence, we extended each primary detection by 2 kb upstream and downstream, forming an extended detection. Following this, we applied CheckV v1.0.1 [47] to further refine our prophage predictions. CheckV not only splits merged prophages within extended regions but also trims over-predicted regions while estimating prophage completeness and confidence. To account for a potential loss of receptor-binding proteins during this trimming, we again extended the CheckV-defined prophage boundaries by 2 kb upstream and downstream. We retained only prophages with a completeness of at least 50%, confidence rated as low, medium or high, and a minimum length of 2kb. This resulted in a dataset of 8,105 prophages from the GWAS dataset. The mean number of detected prophages per genome was 3.2, with a maximum of 14 prophages found in a single genome. Of those, 4,598 prophages were of high-completeness (⩾90%), which yields 1.8 high-quality prophages per isolate. We did not detect any prophages in 49 bacterial genomes. The prophage detection workflow is implemented in Snakemake v8.10.0 and is accessible on GitHub at https://github.com/bioinf-mcb/mgg_prophages.

**Functional annotation.** To functionally annotate predicted prophages in bacterial genomes, we predicted Open Reading Frames (ORFs) prodigal-gv (v2.11.0) [104] which were clustered MMseqs2 v13.45111 [105] with sensitivity `-s 7.5`, using criteria of at least 80% coverage between query and subject sequences and a minimum of 50% amino acid identity, forming clusters (PC80) to streamline downstream analysis. Next, we aligned these protein clusters against the PHROGs database [48] with HHblits from the HHsuite package version 3 [106] using parameters: `-n 2 -cov 0 -p 0.95`. These alignments were then converted into HMM profiles and queried against the ECOD70 database (Evolutionary Classification of Protein Domains [51]; version F70_20200207, develop263) using HHsearch from HHsuite3, retaining hits with a probability of 70% or higher. Within each PC80, we identified all unique ECOD X-levels, selecting the

highest-scoring hit within each level, and reported up to five distinct ECOD T-levels per PC80 cluster based on the highest probability. Finally, we mapped predicted structural topologies onto the proteins to provide a comprehensive functional annotation. Prophage functional annotation is accessible as Snakemake v 8.10.0 workflow on GitHub repository under: https://github.com/bioinf-mcb/mgg_annotation.

**Genome-wide association studies.** To identify statistical associations between bacterial polysaccharide capsules (determined by the predicted K-locus) and protein clusters found in prophages, we conducted a genome-wide association study (GWAS) independently of the functional annotation. Protein clusters were generated by selecting proteins with a minimum length of 300 amino acids and clustering them using MMseqs. We used six different clustering levels, using combinations of identity thresholds (none, 50%, 80%) and coverage thresholds (50%, 80%), with the 50/50 parameter combination by default (PC50 clusters). Rare variants were filtered out by retaining only protein clusters found in at least three bacterial isolates from two different sequence clusters (SCs), resulting in 1,690 PC50 clusters out of an initial set of 3,745 clusters.

GWAS was run separately for each of the 35 K-loci which were found in at least 10 SCs. For each K-locus, we ran pyseer v1.3.11 [107], which is an implementation of the elastic-net approach [108], which combines a ridge regression and lasso regression, with a parameter alpha which determines the amount of mixing between the two penalties, such that $\alpha = 0$ results in ridge regression and $\alpha = 1$ results in lasso regression. We used SCs as covariates in the model to account for the population structure, under the assumption that capsule-specific infection in prophages is primarily shaped by recent horizontal acquisitions rather than deep shared ancestry; this is supported by evidence that most prophages in bacterial genomes are relatively recent acquisitions [33,109].

Given the biological nature of the polysaccharide-RBP specificity problem, we a priori expected that a given K-type would be recognised by a single enzyme family. In this situation, we would expect the associations to be determined by a small number of predictors (ideally a single protein cluster), a situation better suited for lasso-regression which eliminates most weak predictors. However, we also allowed the possibility of multiple predictors due to the potential complexities of the analysis, for example high diversity and mosaicism of RBPs resulting in multiple sequence clusters weakly correlating with the phenotype – a situation better suited for ridge-regression which allows the presence of multiple statistical predictors. Hence, we ran pyseer in two modes, with the first being lasso-dominated ($\alpha = 0.8$) and the second being ridge-dominated ($\alpha = 0.069$). The inputs to pyseer included: (1) a binary matrix for KL types, indicating the presence or absence of the K-locus in question in all 2,527 bacterial isolates; (2) a binary matrix for the presence or absence of each of the 1,690 prophage PC50 clusters in the same isolates; and (3) a mapping of each bacterial isolate to its SC.

The model outputs a regression coefficient ($\beta$) and p-value for each protein cluster, quantifying its statistical association with the K-locus. P-values were Bonferroni-corrected (p-value of 0.05/$N$, where $N = 1,690$ is the number of PC50 protein clusters). While a high $\beta$ value indicates strong association, visualisation of these associations across the bacterial phylogeny using Phandango v1.3.1 [56] revealed that strong statistical associations do not necessarily translate to accurate predictions, especially in imbalanced datasets where certain K-loci are relatively rare. We therefore considered all protein clusters with $\beta > 0$ and p-value below the significant threshold as predictors, and evaluated their classification performance for each K-locus separately on the full dataset using precision, recall, F1 score, and Matthews correlation coefficient (MCC):

• **Precision** measures specificity:

$$\text{Precision} = \frac{\text{TP}}{\text{TP} + \text{FP}}$$

- **Recall** measures sensitivity:

$$\text{Recall} = \frac{\text{TP}}{\text{TP} + \text{FN}}$$

where:

- **True Positive (TP)**: Protein variant present in an isolate with the corresponding K-locus.

- **False Positive (FP)**: Protein variant present in an isolate without the corresponding K-locus.

- **False Negative (FN)**: Protein variant absent from an isolate with the corresponding K-locus.

- **True Negative (TN)**: Protein variant absent from an isolate without the corresponding K-locus.

Additional metrics included:

- **F1 score** balances precision and recall:

$$\text{F1} = 2 \times \frac{\text{Precision} \times \text{Recall}}{\text{Precision} + \text{Recall}}$$

- **Matthews Correlation Coefficient (MCC)** provides a comprehensive measure of prediction quality, incorporating all four confusion matrix outcomes (TP, FP, TN, FN):

$$\text{MCC} = \frac{\text{TP} \times \text{TN} - \text{FP} \times \text{FN}}{\sqrt{(\text{TP} + \text{FP})(\text{TP} + \text{FN})(\text{TN} + \text{FP})(\text{TN} + \text{FN})}}$$

Precision was particularly critical because we aimed to identify highly specific enzymes targeting distinct polysaccharides. A high precision maximises reliability of predictions, even if this occasionally reduces recall. To assess statistical uncertainty of precision and recall, we performed $n = 100$ bootstrap replicates for each K-locus, resampling bacterial isolates with replacement to recalculate TP, FP, and FN values. We derived 95% confidence intervals from the distribution quantiles of these bootstrapped estimates.

Finally, for each PC50 hit selected from the GWAS analysis, we predicted the functions of its sequence members by mapping them to their corresponding PC80 clusters, where functional annotations had previously been assigned. Since different sequence members within the same PC50 cluster could be associated with different PC80 clusters, each member might carry distinct ECOD T-level annotations (or combinations thereof, given the multi-domain nature of phage RBPs). To provide a single functional prediction for each PC50 cluster, we selected the most frequent combination of ECOD T-levels among its sequence members. Additionally, we independently checked whether any sequence member in the PC50 cluster was annotated with the *Pectin lyase-like fold* (ECOD T-level: 207.2.1) in any PC80 cluster. If this annotation was present, we labelled the PC50 cluster as a putative pectin lyase fold.

## Experimental approach

**Depolymerase prediction.** All prophages detected in the KASPAH-REF dataset were considered if they were classified as high or medium completeness (CheckV criteria; see above). Predicted prophage proteins were clustered using MMseqs2 with parameters set to 90% bidirectional sequence coverage and 50% sequence identity. Functional annotations of protein clusters were predicted by searching representative cluster sequences against the PHROGs

and UniClust30 databases using HHblits with default parameters (UniRef30 database, E-value cutoff = $10^{-3}$, minimum sequence coverage = 20%). A protein cluster was classified as a putative depolymerase candidate if it met the following criteria: (1) the average length of proteins in the cluster was at least 500 amino acids; (2) the cluster was functionally annotated as either 'tail fiber/fibre' or 'tail spike'; (3) proteins contained predicted enzymatic domains characteristic of polysaccharide-degrading enzymes–specifically lyases (e.g., hyaluronidases, pectin/pectate lyases, alginate lyases, K5 lyases) or hydrolases (e.g., sialidases, rhamnosialidases, levanases, dextranases, or xylanases).

Domain predictions were carried out by comparing representative sequences against multiple databases, as follows: (a) BLASTp searches against the non-redundant protein sequence (nr) database, using standard parameters (expect threshold = 0.05; word size = 5; matrix = BLOSUM62; gap costs: existence = 11, extension = 1; conditional composition score matrix adjustment); (b) HMMER searches against Reference Proteomes (quick search mode; significance thresholds: sequence E-value = 0.01, hit E-value = 0.03); and (c) HHpred searches against the PDB_mmCIF70 database (HHblits MSA generation from UniRef30; maximum MSA iterations = 3; MSA generation E-value cutoff = $10^{-3}$; minimum sequence identity = 0%; minimum coverage = 20%; secondary structure scoring: during alignment; alignment mode: realign with MAC [local:norealign], MAC realignment threshold = 0.3; number of target sequences = 250; minimum probability in hitlist = 20%).

Using a combination of the two approaches, a total of 469 proteins meeting all selection criteria were identified. These proteins were clustered at high coverage using MMseqs2 (80% coverage and 50% identity), obtaining 124 representative protein sequences. These sequences were then structurally modelled using AlphaFold2 [57] in the monomer mode. The resulting models were screened for conserved structural features, with particular emphasis on the presence of a parallel $\beta$-helix fold characteristic of depolymerases. Structural analysis led to the identification of 50 proteins from 32 clusters, with 22 clusters yielding a single representative and 10 clusters yielding multiple representatives to capture sequence and domain-level variation within those groups (altogether 28). The complete list of these proteins is given in S4A Table, and their genomic coordinates are provided in S5 Table.

**Bacterial strains.** *K. pneumoniae* strains used in this study (S2 Table) originate from collections of the Collection de l'Institut Pasteur (CIP), Paris, France, the National Collection of Type Cultures (NCTC), the UK Health Security Agency (UKHSA), the collection of the Department of Pathogen Biology and Immunology (Wrocław, Poland), and the *Klebsiella* Acquisition Surveillance Project at Alfred Health (KASPAH) from Melbourne, Australia [39,41,42]. Bacteria were grown in Tryptone Soya Broth or Agar (TSB or TSA, Oxoid, Thermo Fisher Scientific, Waltham, MA, USA) at 37°C. The collection includes 128 strains representing 118 distinct serotypes/K-loci.

The *Escherichia coli* strains (Thermo Fisher Scientific, Waltham, MA, USA) were used for recombinant protein production: One ShotTM TOP10 for cloning and One ShotTM BL21 StarTM (DE3) for recombinant protein expression. They were cultured in LB broth, Miller (Luria-Bertani) (Difco, Becton, Dickinson and Company (BD), Franklin Lakes, NJ, USA) at 37 °C supplemented with 100 µg/mL ampicillin (IBI Scientific, Dubuque, IA, USA).

All bacteria were stored at −70°C in a 20% glycerol-supplemented Trypticase Soy Broth (TSB, Becton Dickinson, and Company, Cockeysville, Md, USA).

**Cloning, expression and purification.** The genomic DNA from bacterial strains containing prophages encoding putative depolymerase was extracted using a Pure Link Genomic DNA Mini Kit (Thermo Fisher Scientific, Waltham, MA, USA) according to the manufacturer's protocol. The genes were amplified by PCR using the specific primer pairs (Genomed, Warsaw, Poland) and Platinum SuperFi DNA polymerase (Thermo Fisher Scientific, Waltham, MA, USA). The desired products were subsequently cloned into the Champion pET101 expression vector (Thermo Fisher Scientific, Waltham, MA, USA) with a C-terminal His tag (6x) and propagated in *E. coli* One ShotTM TOP10 strain according to the manufacturer's protocol. Plasmid DNA was then purified and the accuracy of the clones was confirmed by sequencing (Genomed, Warsaw, Poland).

Following the transformation, recombinant depolymerases were expressed in *E. coli* One Shot BL21 Star (DE3) (Thermo Fisher Scientific, USA). Cultures were grown in 500 mL LB broth supplemented with 100 µ g/mL ampicillin at 37 °C.

In addition to the standard expression conditions (1.0 mM IPTG, 18 °C, $OD_{600}$ ≈ 0.5, 16 h) – first attempt, an alternative induction strategy was tested (0.5 mM IPTG, 15 °C, $OD_{600}$ ≈ 1.0, 16 h expression) – second attempt. Cultures were harvested by centrifugation (8,000 × g, 15 min, 4 °C), and pellets were resuspended in 15 mL of lysis buffer (500 mM NaCl, 20 mM $NaH_2PO_4$, pH 7.4). Cells were lysed using three freeze–thaw cycles followed by sonication, and crude lysates were clarified by centrifugation (16,000 × g, 30 min, 4 °C) and filtration through 0.22 µ m filters (Sarstedt, Nümbrecht, Germany). Purification of His-tagged proteins was performed using Dynabeads His-Tag Isolation and Pulldown (Thermo Fisher Scientific, USA) following the manufacturer's instructions. The quality of the purified proteins was analysed by a sodium dodecyl sulfate–polyacrylamide gel electrophoresis (SDS-PAGE) using a 12% polyacrylamide gel and Precision Plus Protein All Blue molecular weight standard (Bio-Rad, Hercules, CA, USA).

For all expression conditions, enzymatic activity was evaluated using standard spot assays performed on both crude lysates and purified His-tagged proteins. Crude lysates were tested directly to enable the detection of activity even at very low expression levels. As a negative control, crude lysates from bacteria carrying the empty Champion pET101 vector were included in all assays to ensure that halo formation did not originate from host-derived or vector-associated background activity.

In addition to employing standard cloning and expression techniques, the production of selected depolymerases (S4A Table) was outsourced to a commercial service provider (GenScript, China). The genes encoding all the aforementioned proteins were synthesised with codon optimisation for expression in *E. coli*, cloned into pET30a expression vector, and prepared by the company as highly purified protein preparations, each bearing an N-terminal 6×His tag.

**Enzymatic functionality assays.** Enzymatic activity was assessed using a spot assay on *K. pneumoniae* serotype collections listed in S2 Table. Bacterial cultures grew to an optical density at 600nm ≈ 1.0 and 0.5 mL was spread evenly onto TSA plates. After drying, 10 µL of each recombinant enzyme was spotted on the bacterial lawn. As a negative control, 10 µL of a sample obtained from the expression and purification of an empty vector–processed using the same protocol as for the recombinant proteins was also applied. Following overnight incubation at 37 °C and an additional 24 h incubation at room temperature, plates were examined for the presence of halo zones. The minimum concentration of protein required to produce a detectable halo on the bacterial lawn (minimal halo-forming concentration; MHFC) was established using the method described previously [110].

**Esterase activity assay for SGNH-domain proteins.** SGNH hydrolases are broadly active on simple acetate esters, and hydrolysis of *p*-nitrophenyl acetate (pNPA) is widely used as a diagnostic assay for SGNH-family esterases, including polysaccharide O-acetylesterases [59–61,111]. Hydrolysis of pNPA yields *p*-nitrophenol, which can be monitored spectrophotometrically, and acetic acid, which produces a detectable pH shift using phenol red [112,113].

**Protein samples.** Two recombinant tail fibres predicted to contain SGNH hydrolase domains (164_08, obtained based on prediction for KL2, and 174_38, obtained based on prediction against KL55) and one depolymerase lacking an SGNH domain (184_43-KL111; negative control) were obtained as highly purified preparations from GenScript (Piscataway, NJ, USA). Proteins were stored at –80 ° C and diluted in 10 mM Tris–HCl (pH 7.5) immediately before use.

**Spectrophotometric pNPA hydrolysis assay.** Enzymatic activity was evaluated in 96-well flat-bottom plates using 4 *p*-nitrophenyl acetate (4p-NPA; Sigma-Aldrich) as a chromogenic substrate. Each reaction well contained a final volume of 200 µL composed of 10 mM Tris-HCl buffer (pH 7.5), protein at final amount of 10 µg, and p-NPA at a final concentration of 100 µM. All Reactions were incubated at 25 ° C, and the release of *p*-nitrophenol was monitored kinetically at 405 nm for 20 min, with absorbance recorded every minute on a microplate spectrophotometer Varioskan LUX (ThermoFisher Scientific). The colour change to yellow, indicating the release of nitrophenol, was also observed visually after incubation. For controls, wells containing substrate without protein (to correct for spontaneous hydrolysis of p-NPA), substrate with the KL111 depolymerase and KL2 and KL55 SGNH hydrolases without substrate were included. For "substrate only" and "substrate + protein" conditions, the mean ± SD was calculated from all six measurements (2 biological × 3 technical replicates). For "protein only" controls, technical triplicates were first averaged within each biological replicate; the mean

± SD was then calculated from $n = 6$ biological replicates (two per protein). All values obtained during the experiments are provided in S3 Data.

**Phenol-red pH-shift assay.** Immediately after completion of the 20-min spectrophotometric measurement, a secondary colorimetric assay was performed to assess acetic acid release during hydrolysis. Phenol red was added to each well to a final concentration of 0.002% (w/v). The pH-dependent colour change of phenol red was then observed visually as a qualitative indicator of hydrolytic activity and measured at 560nm. For "substrate only" and "substrate + protein" conditions, the mean ± SD was calculated from all six measurements (2 biological × 3 technical replicates). For "protein only" controls, technical triplicates were first averaged within each biological replicate; the mean ± SD was then calculated from $n = 6$ biological replicates (two per protein). All values obtained during the experiments are provided in S3 Data.

## Supporting information

**S1 Fig. (Top) Distribution of prophage completeness as estimated by CheckV per K-type for the 35 K-types shown in Fig 1.** Each point shows the completeness of each prophage detected in an isolate with the corresponding K-type. Boxplots are overlaid to show the median values and quantiles. (Bottom) The same but shown for a single representative per phage variant with the highest value of completeness. The data underlying this Figure can be found at Figshare (https://doi.org/10.6084/m9.figshare.29181188), and can be reproduced using code archived in Zenodo (https://doi.org/10.5281/zenodo.18699826).
(PDF)

**S2 Fig. (Left) Full distribution of the number of Sequence Clusters (SCs) per K-type for all K-types in the dataset from Fig 1.** (Right) Full distribution of the number of phage variants, obtained at wGRR = 0.95, per K-type for all K-types in the dataset. The data underlying this Figure can be found at Figshare (https://doi.org/10.6084/m9.figshare.29181188), and can be reproduced using code archived in Zenodo (https://doi.org/10.5281/zenodo.18699826).
(PDF)

**S3 Fig. Relationship between Precision and Recall for the 35 genetic variants from Fig 2A, i.e., those with F1 ≥ 0.5 and MCC ≥ 0.5.** The black line shows the best-fit linear model (Pearson correlation) to these data. Colours indicate the corresponding functional predictions obtained using PHROGs. The Pearson correlation coefficient and associated p-value are shown in the plot. The data underlying this Figure can be found at Figshare (https://doi.org/10.6084/m9.figshare.29181188), and can be reproduced using code archived in Zenodo (https://doi.org/10.5281/zenodo.18699826).
(PDF)

**S4 Fig. Each row represents one of the 14 recombinant proteins from Fig 4 that showed enzymatic activity against a panel of 118 reference K-loci.** Columns indicate K-loci corresponding to the bacterial hosts of the prophages from which the proteins originated. Green cells denote K-loci on which the enzyme was active; pink cells indicate the K-locus of the host from which the enzyme's prophage was derived. Rows with only green cells indicate activity against the host's own K-locus, while rows with both pink and green highlight activity on a different K-locus than the prophage's host. The data underlying this Figure can be found at Figshare (https://doi.org/10.6084/m9.figshare.29181188), S4 Table, and can be reproduced using code archived in Zenodo (https://doi.org/10.5281/zenodo.18699826).
(PDF)

**S5 Fig. Only a fraction of predicted depolymerases can be overexpressed.** Analysis of recombinant depolymerases from connected component 1 in the sequence similarity network (manuscript Fig 4) that did not overexpress. (A) Connected component 1 from the sequence similarity network of recombinant and predicted depolymerases (B) AlphaFold3 homotrimer models of proteins from A which were active or did not overexpress. (C) CheckV and Kaptive metadata for

prophages and bacterial K-loci from which the genes encoding putative depolymerase were cloned. The data underlying this Figure can be found at Figshare (https://doi.org/10.6084/m9.figshare.29181188), S2 Data, S4 Table, and can be reproduced using code archived in Zenodo (https://doi.org/10.5281/zenodo.18699826).
(PNG)

**S6 Fig. Only a fraction of predicted depolymerases can be overexpressed.** Analysis of recombinant depolymerases from connected component 4 in the sequence similarity network (manuscript Fig 4) that did not overexpress. (A) Connected component 4 from the sequence similarity network of recombinant and predicted depolymerases. (B) AlphaFold3 homotrimer models of proteins from A which were active or did not overexpress. (C) CheckV and Kaptive metadata for prophages and bacterial K-loci from which the genes encoding putative depolymerase were cloned. The data underlying this Figure can be found at Figshare (https://doi.org/10.6084/m9.figshare.29181188), S2 Data, S1 Table, S4 Table, and can be reproduced using code archived in Zenodo (https://doi.org/10.5281/zenodo.18699826).
(PNG)

**S7 Fig. Comparison of prophage quality and genomic features between active and inactive depolymerases.** We tested whether the lack of detectable activity or expression could be explained by the genomic context of the corresponding prophages. Each depolymerase was classified as expressed and active (dark red) or not expressed or not active (light blue), and the prophage of origin was analysed for several genomic attributes. (A) Prophage completeness estimated by CheckV. (B) Number of *K. pneumoniae* sequence types (STs) containing closely related prophages, used as a proxy for prophage age and vertical inheritance. (C) Size of the prophage cluster (number of similar prophages in the dataset), reflecting the frequency of the element across hosts. (D) Number of annotated transposases per prophage genome, a hallmark of prophage degradation. All tested prophages were complete or near-complete, and no significant differences were observed between active and inactive groups for any feature (Fisher's exact and Wilcoxon tests, $p > 0.1$). These analyses indicate that inactivity or low expression cannot be explained by prophage degradation or domestication. The data underlying this Figure can be found at Figshare (https://doi.org/10.6084/m9.figshare.29181188), S4 Table, S5 Table, and can be reproduced using code archived in Zenodo (https://doi.org/10.5281/zenodo.18699826).
(PNG)

**S8 Fig. Prophages carrying multiple depolymerases from the manual search.** (A) Prophage tail gene modules from which 2 putative depolymerase genes were cloned and tested along with their AlphaFold3 homotrimer models. Colours correspond to active depolymerases (pink), putative depolymerases which did not overexpress (grey), central tail fiber (violet), lysozyme (red), resolvase (orange). (B) CheckV and Kaptive metadata for prophages and bacterial K-loci loci from which the genes encoding putative depolymerase were cloned. The data underlying this Figure can be found at Figshare (https://doi.org/10.6084/m9.figshare.29181188), S2 Data, S4 Table, S5 Table, and can be reproduced using code archived in Zenodo (https://doi.org/10.5281/zenodo.18699826).
(PNG)

**S9 Fig. Similarity of active and not produced depolymerases.** Two examples of protein pairs selected for recombinant overexpression and enzymatic activity testing (A, B). For each pair, the receptor-binding domains share >97% percentage identity in amino-acid sequence, and the corresponding AlphaFold3 homotrimer models are shown. Amino-acid substitutions between proteins in a pair are mapped onto the models and displayed as yellow/orange balls (on each monomer), showing their distribution across multiple parts of the structures. The data underlying this Figure can be found at Figshare (https://doi.org/10.6084/m9.figshare.29181188), S4 Table, and can be reproduced using code archived in Zenodo (https://doi.org/10.5281/zenodo.18699826).
(PNG)

**S10 Fig. Upset plot showing the number of high-quality (at least 99% complete) prophages encoding each ECOD domain or combination of ECOD domains shown in** Fig 6A **within PHROG-defined RBPs, stratified by K-locus.** The most frequent multi-domain combinations are gp11/gp12 + SGNH hydrolase and pectin-lyase-like + Orf210 N-terminal or gp11/gp12 domains. The data underlying this Figure can be found at Figshare (https://doi.org/10.6084/m9.figshare.29181188), and can be reproduced using code archived in Zenodo (https://doi.org/10.5281/zenodo.18699826).
(JPG)

**S11 Fig. Number of high-quality (at least 99% complete) prophages whose RBPs lack any of the domains shown in** Fig 5A **but encode each ECOD domain up to 2 protein-coding genes upstream or downstream from the protein annotated as RBP.** The data underlying this Figure can be found at Figshare (https://doi.org/10.6084/m9.figshare.29181188), and can be reproduced using code archived in Zenodo (https://doi.org/10.5281/zenodo.18699826).
(JPG)

**S12 Fig. Tail regions of representative 16 prophages found in** *K. pneumoniae* **isolates with the KL1 locus.** These prophages were chosen from all high-quality (100% completeness and 'high' confidence) prophages in KL1 isolates by accounting for bacterial lineage (SC) and phage variant (wGRR = 0.5 threshold). For clarity, prophage genomes were manually curated to highlight tail gene clusters by removing genes located downstream of the tail length tape measure protein. For two prophages (KVV_B1_PHAGE002_M and KQS_B1_PHAGE024_M) genes upstream of the tail tube protein were also removed. Proteins were annotated by the ECOD with a e-value of. Tail regions were compared using Clinker, and pairwise protein identity across their full lengths is indicated by the greyscale shown in the colour bar. The data underlying this Figure can be found at Figshare (https://doi.org/10.6084/m9.figshare.29181188), and can be reproduced using code archived in Zenodo (https://doi.org/10.5281/zenodo.18699826).
(PDF)

**S13 Fig. Number of high-quality (at least 99% complete) prophages that encode acetyltransferases as detected by PHROG (hit to 'acetyltransferase' or 'O-acetyltransferase') with and, or by ECOD (hit to any ECOD containing a string acetyltransf with and.** Yellow shows hits to ECOD-only, orange shows hits to PHROG only, red shows hits to both and grey shows this to neither database. The data underlying this Figure can be found at Figshare (https://doi.org/10.6084/m9.figshare.29181188), and can be reproduced using code archived in Zenodo (https://doi.org/10.5281/zenodo.18699826).
(JPG)

**S14 Fig. Global distribution of** *K. pneumoniae* **sequence types (STs) across the GWAS, KlebNNSsero and Pathogenwatch datasets.** (A) Overlap in *K. pneumoniae* STs between the three datasets, showing shared and dataset-unique ST subsets. (B) Jitter plot with boxplots showing, for each of the 405 shared STs, the number of countries in which the ST has been detected (Y-axis) plotted against the number of continents where it occurs (X-axis). (C) Stacked bar plots illustrating the country and continent spread of STs within each subset, normalised by the total number of STs in that subset. (D) Bar plot showing the number of sequence types (STs; $n = 542$) in the GWAS dataset (y-axis) as a function of the number of countries in which they have been reported (x-axis), excluding the two GWAS source countries (Italy and Australia). (E) Analogous bar plot showing the number of STs as a function of the number of continents in which they have been reported, excluding the two GWAS source continents (Europe and Oceania). Across the 542 STs in the GWAS dataset, 390 STs (72%) have been reported in at least two countries and 314 STs (58%) in at least three countries, while 347 STs (64%) have been reported on at least two continents. Notably, 382 STs (70%) were detected in at least one additional country beyond Italy and Australia, and 311 STs (57%) were detected on at least one additional continent beyond Europe and Oceania. Together, these results indicate that the majority of lineages represented in the GWAS dataset are globally

distributed. The data underlying this Figure can be found at Figshare (https://doi.org/10.6084/m9.figshare.29181188), S1 Data, and can be reproduced using code archived in Zenodo (https://doi.org/10.5281/zenodo.18699826).
(PNG)

**S15 Fig. Effect of low-quality prophages on GWAS predictions.** Low-quality prophages were defined as CheckV completeness <99% or medium/low CheckV confidence; high-quality prophages had completeness 99% and high confidence. (A) Scatter plots compare, for 35 K-loci, the top F1 scores obtained from GWAS runs on 2 prophage datasets, proteins from high- and low-quality prophages (X-axis) versus proteins from high-quality prophage only (Y-axis), and six protein clustering settings. Each point represents a K-locus; row and column labels correspond to the parameters used for prophage protein clustering – i.e., bidirectional protein sequence coverage (column label 'coverage') and protein amino-acid sequence identity (row label 'identity'); outliers with F1 outside the interquartile range are labelled; dashed lines indicate the diagonal; Pearson correlation coefficients (r) are shown in the top-left corner of each panel. (B) Bar plots show, for each K-locus (X-axis), the top F1 score (Y-axis) across the same six clustering settings, comparing top F1-scores obtained from GWAS run on proteins from high- and low- quality prophages (orange) versus high-quality prophages only (blue). Top GWAS predictors from two prophage datasets show overall positive correlation across all clustering. The exclusion of low-quality prophages for most outliers leads to decrease in F1-score value (e.g., KL127). The data underlying this Figure can be found at Figshare (https://doi.org/10.6084/m9.figshare.29181188), and can be reproduced using code archived in Zenodo (https://doi.org/10.5281/zenodo.18699826).
(PNG)

**S1 Table. List of 58 active depolymerases from previously published studies on virulent phages, based on Cheetham and colleagues [27].**
(XLSX)

**S2 Table. List of 128 *K. pneumoniae* strains representing the reference collection of 118 K-types used in this study.**
(XLSX)

**S3 Table. List of GWAS-based depolymerase predictions, constituting 26 protein sequences selected based on a manually-curated approach described in S1 Text, including 12 protein sequences classified as 'strong' predictions and 14 protein sequences classified as 'likely' predictions.**
(XLSX)

**S4 Table. (A) List of experimentally tested, recombinantly produced proteins, (B) Optimisation of expression and activity for eight selected recombinant proteins, (C) List of expression conditions for *Klebsiella* phage depolymerases used in the literature.**
(XLSX)

**S5 Table. Map of proteins from S4 Table onto prophages from S2 Data.**
(XLSX)

**S1 Text. GWAS-based depolymerase predictions.**
(PDF)

**S2 Text. Experimental validation of capsule depolymerase activity.**
(PDF)

**S3 Text. Sequence and structure comparison of predicted and verified depolymerases.**
(PDF)

**S1 Data. Metadata table for all input 3,911 *Klebsiella* sp. isolates, including genome and assembly identifiers, collection source, species, sequence type (ST), K- and O-locus assignments with confidence metrics, assembly statistics and GWAS inclusion flag.**
(XLSX)

**S2 Data. Metadata table for all 8,105 prophages detected across the 2,527 *Klebsiella pneumoniae* species complex isolates in the GWAS dataset, including prophage and contig identifiers, genomic coordinates, length, CheckV completeness and confidence estimates, bacterial host metadata (species, ST, K- and O-locus), and phage variant cluster assignments at multiple wGRR thresholds.**
(XLSX)

**S3 Data. Raw absorbance measurements from the SGNH esterase activity assays.** Sheet S3A contains time-course absorbance readings at 405 nm (pNPA hydrolysis assay) across biological and technical replicates. Sheet S3B contains end-point absorbance readings at 560 nm (phenol-red pH-shift assay).
(XLSX)

## Author contributions

**Conceptualization:** Zuzanna Drulis-Kawa, Rafal Mostowy.

**Data curation:** Janusz Koszucki, Jade Leconte.

**Formal analysis:** Aleksandra Otwinowska, Janusz Koszucki, Vyshakh R. Panicker, Jade Leconte, Sebastian Olejniczak, Bogna Smug, Rafal Mostowy.

**Funding acquisition:** Rafal Mostowy.

**Investigation:** Aleksandra Otwinowska, Janusz Koszucki, Vyshakh R. Panicker, Eduardo P. C. Rocha, Bogna Smug, Barbara Maciejewska, Zuzanna Drulis-Kawa, Rafal Mostowy.

**Methodology:** Janusz Koszucki, Barbara Maciejewska, Zuzanna Drulis-Kawa, Rafal Mostowy.

**Project administration:** Barbara Maciejewska, Zuzanna Drulis-Kawa, Rafal Mostowy.

**Resources:** Kathryn E. Holt, Edward J. Feil, Zuzanna Drulis-Kawa, Rafal Mostowy.

**Software:** Janusz Koszucki.

**Supervision:** Barbara Maciejewska, Zuzanna Drulis-Kawa, Rafal Mostowy.

**Validation:** Aleksandra Otwinowska, Janusz Koszucki, Bogna Smug, Rafal Mostowy.

**Visualization:** Aleksandra Otwinowska, Janusz Koszucki, Bogna Smug, Rafal Mostowy.

**Writing – original draft:** Bogna Smug, Zuzanna Drulis-Kawa, Rafal Mostowy.

**Writing – review & editing:** Aleksandra Otwinowska, Janusz Koszucki, Vyshakh R. Panicker, Jade Leconte, Sebastian Olejniczak, Kathryn E. Holt, Edward J. Feil, Eduardo P. C. Rocha, Bogna Smug, Barbara Maciejewska, Zuzanna Drulis-Kawa, Rafal Mostowy.

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
