## [Editor Report · Decision Letter 0]

22 Jul 2025

Dear Dr Mostowy,

Thank you for submitting your manuscript entitled "Capsular specificity in temperate phages of Klebsiella pneumoniae is driven by diverse receptor-binding enzymes" for consideration as a Research Article by PLOS Biology.

Your manuscript has now been evaluated by the PLOS Biology editorial staff, as well as by an academic editor with relevant expertise, and I am writing to let you know that we would like to send your submission out for external peer review.

Once your full submission is complete, your paper will undergo a series of checks in preparation for peer review. After your manuscript has passed the checks it will be sent out for review. To provide the metadata for your submission, please Login to Editorial Manager (https://www.editorialmanager.com/pbiology) within two working days, i.e. by Jul 24 2025 11:59PM.

Kind regards,

Melissa

Melissa Vazquez Hernandez, Ph.D.

Associate Editor

PLOS Biology

---

## [Decision Letter · Decision Letter 1]

9 Sep 2025

Dear Dr Mostowy,

Thank you for your patience while your manuscript "Capsular specificity in temperate phages of Klebsiella pneumoniae is driven by diverse receptor-binding enzymes" was peer-reviewed at PLOS Biology. Your manuscript has been evaluated by the PLOS Biology editors, an Academic Editor with relevant expertise, and by three independent reviewers. I sincerely apologize for the incredibly long review process.

As you will see in the reviewer reports, although the reviewers acknowledge the potential interest in your findings, they have also raised a substantial number of crucial concerns. Based on their specific comments and following discussion with the Academic Editor, it is clear that a substantial amount of work would be required to meet the criteria for publication in PLOS Biology. Given our and the reviewer interest in your study, we would be open to inviting a comprehensive revision of the work; however, this would need to thoroughly address all the reviewers' comments in full. A successful revision will need to provide functional validation and address the concerns related to the expression systems, as well as be more clear about the relevance and novelty of the study. Furthermore, the revision must also address the mismatch between the GWAS analysis and the protein expression.

Given the extent of revision that would be needed, we cannot make a decision about publication until we have seen the revised manuscript and your response to the reviewers' comments. Your revised manuscript would need to be seen by the reviewers again, but please note that we would not engage them unless their main concerns have been addressed.

We appreciate that these requests represent a great deal of extra work, and we are willing to relax our standard revision time to allow you 6 months to revise your study. Please email us (plosbiology@plos.org) if you have any questions or concerns, or envision needing a (short) extension.

**IMPORTANT - SUBMITTING YOUR REVISION**

*Resubmission Checklist*

*Published Peer Review*

*PLOS Data Policy*

*Blot and Gel Data Policy*

Sincerely,

Melissa

Melissa Vazquez Hernandez, Ph.D.

Associate Editor

PLOS Biology

REVIEWERS' COMMENTS:

Reviewer #1:

The study by Owtinowska and colleagues uses a genome-wide association approach with experimental validation to identify prophage-encoded proteins with specificity for and activity against Klebsiella pneumoniae capsular loci. The authors identify prophage loci and protein clusters that are significantly associated with specific loci. They also find that protein clusters that are strong predictors of capsule (K) type have a relatively high correlation with functional depolymerase activity against that capsule type. They also explore the types of prophage proteins associated with capsule types and find that SGNH hydrolases are found frequently, and appear to be found in a greater diversity of K types, suggesting they may be important mediators of phage infection. The technical merit of the study is high, with a large number of genomes used from two different collections, and rigorous genomics, statistical, and experimental approaches used. The experimental design is logical and rigorous, including the synthesis and testing of a large number of prophage proteins. As written, the novelty and significance of the findings is less clear.

Major Comments:

Based on the introduction, it is somewhat unclear what the significance of the study is and what the important gap in knowledge is that this study will fill. Was the goal to identify novel proteins that may target the capsule for degradation, perhaps as eventual therapeutics or bioremediation to control Klebsiella populations? Is the focus on integrated prophages a way to efficiently screen for these proteins, as opposed to physical searches for lytic phages in the environment? The gap in knowledge needs to be more explicitly stated.

Correspondingly, the significance of the results needs to be conveyed relative to the gap in knowledge. It seems that the takeaway is that there was some success in predicting and validating capsule depolymerases, but that often the proteins in prophages are inactive. Does the identification of these new depolymerases drive the field forward? In addition, there are other categories of phage proteins that may be associated with capsule type and infectivity. These SGNH hydrolases and their potential deacetylase activity is discussed over several paragraphs in the discussion, but these proteins are not experimentally tested and make up a small portion of the results. Overall, I think the discussion needs to convey the broad importance of the findings.

Minor Comments:

Figure 2A: This is shaded by species but not sequence cluster, contradicting the figure legend.

Figure 2B: It would be more helpful to the reader if this showed only the K types included in the analysis, whereas Figure S2 can show all the K types considered.

Page 9, lines 16-20: I would suggest moving a brief explanation of precision and recall, F1 and Matthews here, and how they all use TP, FP, TN and FN values for their calculation. This context is important for interpreting the results and figures.

Reviewer #2:

This manuscript presents a comprehensive investigation into the genetic determinants of capsule tropism in temperate phages of K. pneumoniae. The authors analyzed 3,900 Klebsiella genomes (spanning diverse ecological niches) and applied GWAS to correlate prophage protein clusters (from 8,105 prophages) with bacterial K-loci. Furthermore, a total of 60 putative depolymerases were expressed as recombinant proteins, and enzymatic activity was tested on a reference panel of 119 K-types.

The results showed that most predicted depolymerases (34/60) failed to yield soluble products, and 6 lacked activity against the tested strains. Besides, 5 of 14 enzymes displayed activity against K-types different from those of the prophage host. The authors further proposed that SGNH-domain-containing RBPs might rely on capsule acetylation and that deacetylation could represent a strategy used by temperate phages to recognize and initiate infection.

While the work is ambitious and technically demanding, many conclusions are insufficiently supported by experimental evidence. In particular, most predicted depolymerases failed to yield soluble or active proteins, and proposed roles of SGNH hydrolases in capsule deacetylation remain speculative without functional validation. Overall, the manuscript reports substantial effort but provides limited novel biological insights, raising concerns about its suitability for PLOS Biology.

Major Comments

1. Generality of Study Population: The study population was derived only from Australia (clinical isolates) and Italy (diverse sources). The authors should discuss the implications of this limited sampling for the global diversity of Klebsiella prophages and RBPs.

2. Validation of SGNH Hydrolase Function: The proposed O-deacetylation activity of SGNH domain proteins is not experimentally demonstrated. Functional assays are needed to support this claim.

3. Protein Expression and Activity: It is premature to conclude that non-expressed proteins are non-functional (p. 24, lines 26-28), particularly as the expression system used failed to yield soluble products for the majority of candidates. Expression vectors, tags, or conditions may affect protein folding and activity. The authors should consider optimizing recombinant expression strategies.

4. Novelty and Impact - Although correlations between prophages and K-loci were examined, most depolymerase predictions remain unvalidated, limiting the study's contribution to understanding capsule specificity.

Minor Comments

1. Page 10, line 13: Define CATH, PDB, and AFDB at first mention.

2. Page 10, line 24: Remove "and."

3. Page 13, line 28: Correct "brea"

4. Page 13, line 29: remove duplicate "the."

Reviewer #3:

The authors combine large Klebsiella genomic databases to develop a genome wide association study between K-loci and phage proteins. They first mine Klebsiella genomes for prophages, extract these and cluster the proteins, then use these as input for the GWAS, with appropriate controls for the underlying population structure. In parallel they perform functional validation of various candidates from one of the collections. Importantly, they find that very few of the predicted proteins confer the expected functionality.

Overall the analysis is thorough, the manuscript well-written throughout and the results support the conclusions. The choice of datasets is rational and encompasses a range of ecologies. The key result (for me) is that so few of the predicted receptor-binding enzymes appear to be functional in the screening assay. I agree with the authors that this has important consequences for mining prophages for functional enzymes.

I felt there was a slight mismatch between the GWAS analysis and the recombinant expression part of the project. GWAS can identify associations without a reliance on homology (as noted by the authors), but how were the candidate recombinant RBPs found from the prophage genomes? On page 13, line 16 this is simply referred to as 'meeting initial selection criteria' and is presumably homology based? I'd suggest reiterating those criteria here. The authors do a decent of job of aligning the results from these parallel arms of the project in the results, but the introduction could be a bit clearer.

The authors mention degraded / cryptic prophages in the discussion but should consider whether/how these would influence the GWAS? My thinking here is that once non-functional there will be no selection on these genes, which could affect the GWAS. I believe the use of different clustering levels (50%, 80%) would mitigate this effect, but the authors could discuss this.

Minor points:

Suggest giving an overview of the use of elastic-net regression models and their usage for GWAS. I believe this is an appropriate technique, but an overview would be helpful for non popgen familiar readers.

As noted in the discussion, the halo assay is quite limited for inferring activity. Are there other methods that could be used for future work?

Page 2, line 13: 'great model'- slightly colloquial.

Typo, page 26, line 6. 'quasipenumoniae'

Page 26. Line 10- 'all isolates'.

Page 27. Line 19- Code snippets- unfinished sentence. Link to GitHub repo perhaps.

---

## [Decision Letter · Decision Letter 2]

4 Feb 2026

Dear Dr Mostowy,

Thank you for your patience while we considered your revised manuscript "Capsular specificity in temperate phages of Klebsiella pneumoniae is driven by diverse receptor-binding enzymes" for consideration as a Research Article at PLOS Biology. Your revised study has now been evaluated by the PLOS Biology editors, the Academic Editor and the original reviewers.

In light of the reviews, which you will find at the end of this email, we are pleased to offer you the opportunity to address the remaining points from the reviewers in a revision that we anticipate should not take you very long. Specifically, we require clarification on the reproducibility on Fig 7 as highlighted by R1, as well as further context on how the findings will help make more effective phage therapeutics and capsule-based vaccines. To address R1 and R2, please be clear on the advance of the study, but be careful with overstatements. We will then assess your revised manuscript and your response to the reviewers' comments with our Academic Editor aiming to avoid further rounds of peer-review, although we might need to consult with the reviewers, depending on the nature of the revisions.

Additionally, please address the following editorial requests:

1) Please add the weblink of the funding agencies in the Financial Disclosure statement in the manuscript details during submission and in the manuscript.

Please supply the numerical values either in the a supplementary file or as a permanent DOI’d deposition for the following figures:

Figure 2B, 3ABC, 6AB, 7BC, S1, S2, S3, S4, S7A-D, S12, S14, S15B-E, S16AB

*I am aware that the scripts to reproduce Figs 1-7 are on Github, but it seems that those for supplementary figures are missing?

*Thank you for providing the scripts in GitHub. However, because Github depositions can be readily changed or deleted, please make a permanent DOI’d copy (e.g. in Zenodo) and provide this URL in the manuscript and Data Availability Statement.

3) Please cite the location of the data clearly in all relevant main and supplementary Figure legends, e.g. “The data underlying this Figure can be found in S1 Data” or “The data underlying this Figure can be found in https://doi.org/10.5281/zenodo.XXXXX”

4) Please provide the tree files for the phylogenetic trees in Figures 2A. Please make sure all relevant figures have scale bars.

5) Supplementary files (e.g., excel). Please ensure that all data files are uploaded as 'Supporting Information' and are invariably referred to (in the manuscript, figure legends, and the Description field when uploading your files) using the following format verbatim: S1 Data, S2 Data, etc. Multiple panels of a single or even several figures can be included as multiple sheets in one excel file that is saved using exactly the following convention: S1_Data.xlsx (using an underscore).

6) Please ensure that your Data Statement in the submission system accurately describes where your data can be found and is in final format, as it will be published as written there

**IMPORTANT - SUBMITTING YOUR REVISION**

*Resubmission Checklist*

*Published Peer Review*

*PLOS Data Policy*

*Blot and Gel Data Policy*

Sincerely,

Melissa

Melissa Vazquez Hernandez, Ph.D.

Associate Editor

PLOS Biology

REVIEWERS' COMMENTS

Reviewer #1:

This manuscript by Otwinowska is significantly improved. It is very helpful to lay out the 3 gaps in knowledge in the introduction and return to them in the discussion. This helps frame the article and keep the reader focused on the significant findings. The added functional data on SGNH hydrolase raises the significance of the work, increasing the likelihood that these enzymes enable an alternative infection approach separate from depolymerase activity. The section briefly explaining the metrics of precision and recall is also helpful. I have a few remaining minor comments:

1- Figure 7 photographs do not match the conditions shown in the graphs. This makes their interpretation a bit difficult and raises the concern that some conditions seen in the photograph were not reproducible or were not quantified. It seems like the conditions should match across the graphs and photographs in Figures 7B and C.

2- As a reader, I was still asking myself the question: If we fill the 3 gaps in knowledge, then we can do what? Perhaps it is have a better understanding of the basic biology of temperate phages, and that would be valid but would need some context. But the authors state in the discussion that the findings carry "direct implications for designing more effective phage therapeutics and capsule based vaccines" (Lines 568-570), but I am still not sure how they do that.

Reviewer #2:

In the revised manuscript, the authors have conducted additional experiments, including testing alternative expression conditions for eight representative predicted depolymerases and validating the esterase activity of SGNH-domain RBPs. Additionally, the authors have modified their conclusions/discussion to highlight that capsule specificity in temperate phages involves a diverse set of receptor-binding proteins and that functional activity is difficult to predict from sequence alone.

Overall, the data presented are solid and well-interpreted. The manuscript has been improved by the inclusion of these new results and the refinement of several statements, which is appreciated.

Nevertheless, in my opinion, the current data still do not fully resolve the study's central questions regarding the genetic and functional basis of specificity in temperate phages. Specifically, it remains unclear which prophage genes mediate capsule tropism and what exactly drives capsular specificity in Klebsiella pneumoniae temperate phages. Therefore, despite the revisions, a definitive conclusion regarding the primary driver of specificity is still lacking.

Reviewer #3:

The authors have addressed all of my concerns. I believe the interpretation of results is careful, measured and appropriate. The combination of analysis of a large dataset of systematically collected genomes and high-throughput expression and screening make this paper appropriate for PLoS Biology.

---

## [Editor Report · Decision Letter 3]

4 Mar 2026

Dear Rafal,

Thank you for the submission of your revised Research Article "Capsular specificity in temperate phages of Klebsiella pneumoniae is driven by diverse receptor-binding enzymes" for publication in PLOS Biology. On behalf of my colleagues and the Academic Editor, Britt Koskella, I am pleased to say that we can in principle accept your manuscript for publication, provided you address any remaining formatting and reporting issues. These will be detailed in an email you should receive within 2-3 business days from our colleagues in the journal operations team; no action is required from you until then. Please note that we will not be able to formally accept your manuscript and schedule it for publication until you have completed any requested changes.

PRESS

Sincerely,

Melissa

Melissa Vazquez Hernandez, Ph.D., Ph.D.

Associate Editor

PLOS Biology
